# Serological evidence and factors associated to liver damage in malaria-typhoid infected patients consulting in two health facilities, Yaoundé-Cameroon

Madeleine Yvanna Nyangono Essam[1], Arnaud Fondjo Kouam[1,2]*,
Armelle Gaelle Kwesseu Fepa[1], Armel Jackson Seukep[2], Elisabeth Menkem Zeuko'o[2],
Felicité Syntia Douanla Somene[3], Nembu Erastus Nembo[2], Paul Fewou Moundipa[1],
Frédéric Nico Njayou[1]*

1 Department of Biochemistry, Faculty of Science, University of Yaoundé 1, Yaoundé, Cameroon,
2 Department of Biomedical Sciences, Faculty of Health Sciences, University of Buea, Buea, Cameroon,
3 Department of Nursing Sciences, Faculty of Health Sciences, University of Buea, Buea, Cameroon

* arnaudkouam@yahoo.fr, kouam.fondjo@ubuea.cm (AFK); njayou@yahoo.com (FNN)

## Abstract

Malaria and typhoid fever remain major health issues in developing countries like Cameroon, with frequent co-infections. The potential liver damage caused by their respective causative pathogens is overlooked in health management, posing a significant risk of severe liver injury and worsening patient conditions. Accordingly, this study investigated the risk factors associated with liver damage in febrile patients with malaria, typhoid fever, or co-infection at the Obili District Medical Center and Mvog-Betsi Dominican Hospital Center, Yaoundé, Cameroon. Over 8 months, 350 febrile patients were evaluated for their adherence to preventive measures concerning malaria and typhoid fever using a structured questionnaire. Blood samples were analyzed for *Plasmodium* species and *Salmonella* antibodies and liver enzyme levels were measured. Liver damage was assessed using the Roussel Uclaf Causality Assessment Method. Risk factors for infections and liver damage were identified using Fisher's Exact and Chi-Square tests, with significance set at P < 0.05. Among participants tested, 129 (36.86%) and 106 (30.29%) were positive for malaria and typhoid, respectively, while 56 (16.00%) were co-infected. Men (49.2% and 37.3%) were more affected than women (30.6% and 26.7%) for both malaria and typhoid, respectively. Participants aged between 20 and 40 years were the most affected by malaria (21.4%) and typhoid fever (17.4%). The non-use of mosquito nets, presence of standing water, bushes, and garbage dumps near residences were significant risk factors (Relative risk: RR > 2.1; P < 0.0001) for contracting malaria or typhoid fever. The co-infection status (Chi-2 = 18.30; P < 0.0001), parasite density (Chi-2 = 9.8; P = 0.0074), and delay before consulting (Chi-2 = 13.23; P = 0.0013) were significant risk factors for the occurrence of liver injuries. Our findings demonstrated that

**Data availability statement:** All relevant data are within the manuscript and its Supporting Information files.

**Funding:** The author(s) received no specific funding for this work.

**Competing interests:** The authors have declared that no competing interests exist.

the alteration of liver enzyme activity, reflecting liver damage among patients with malaria, typhoid, or malaria-typhoid co-infection, is a reality and should be considered during the patient treatment.

## Introduction

Malaria and typhoid fever are diseases that can affect the same person at once, persistently remaining endemic, posing a major public health issue as they account for hundreds of thousands of deaths each year despite the efforts made by the World Health Organization (WHO) and the governments in tropical areas, especially in sub-Saharan Africa [1,2].

Malaria, an acute feverish disease spread by the bite of the female Anopheles mosquito, is caused by a protozoan belonging to the genus *Plasmodium* [3]. Five species are capable of infecting humans: *P. malariae*, *P. vivax*, *P. ovale*, *P. knowlesi*, and *P. falciparum*, the latter being the most virulent [4]. In 2022, 249 million cases of malaria and over 600,000 deaths were reported [5]. Sub-Saharan Africa accounted roughly 93.6% (233 million) of the incidents and 95.4% (580,000) of the fatalities globally. About 80% of all fatalities in this area were noted among children younger than 5 and expectant mothers, rendering them at high-risk demographic [2,5]. Throughout the same timeframe, Cameroon documented close to 6.3 million cases with around 4,000 fatalities, positioning malaria as a primary reason for hospital visits [6].

Similarly, typhoid fever is a severe systemic infection resulting from an entero-bacterium called *Salmonella Enterica Serotype Typhi*. Various subtypes can also contribute to typhoid fever, specifically: *S. para-typhi A*, *S. para-typhi B*, and *S. para-typhi C* [7]. This infection is usually linked to poor socio-economic conditions. Transmission mainly takes place via the fecal-oral route, as well as through the intake of contaminated food and water [8]. The enhancement of living standards and the use of antibiotics have resulted in its elimination in developed nations. Nevertheless, in developing countries, typhoid fever accounts for around 110,000 fatalities annually [1]. In Cameroon, various studies report typhoid fever being commonly diagnosed in healthcare settings, with a prevalence around 50% among patients with fever [9,10].

Although these two infections are transmitted by distinct pathogens and through different mechanisms, they both share similar symptoms as well as a mandatory hepatic stage during their infectious cycle in humans [11–14]; which is not without adverse consequences on the liver, an organ performing many functions in the body [6,15,16]. Indeed, *Plasmodium* or *Salmonella* species are capable of invading liver cells, causing congestion of the organ as well as inflammation followed by cellular necrosis [7,17,18]. If treatment is not initiated early, or is poorly administered, malaria and typhoid fever can lead to severe liver damage [15,19,20].

Regrettably, in areas where malaria and typhoid fever are common, their adverse impacts on the liver are not always considered during healthcare

management. Additionally, the population seems not realize that inadequate handling of malaria and/or typhoid fever can cause significant liver injury, posing a risk for liver failure, which may result in the patient's death [21–23]. In light of this situation, one might question what kinds of liver damage are caused by malaria and typhoid fever? What risk behaviors in populations lead to malaria and typhoid fever and subsequently to liver damage in infected patients? To address these questions, the present study aimed to determine the type of liver damage and the associated risk factors in malaria, typhoid fever, or malaria-typhoid co-infected patients among febrile patients consulting at the Obili District Medical Center (CMA) and the Dominican Hospital Center (CHD) Saint Martin de Porres in Mvog-Betsi, Yaoundé, Cameroon.

## Materials and methods

### Description of the study area

This study was conducted in two hospitals in the city of Yaoundé, specifically the District Medical Center (CMA) of Obili and the Saint Martin de Pores Dominican Hospital (CHD) of Mvog-Betsi, situated in the Yaoundé III and VI municipalities, respectively. Yaoundé serves as the Political Capital of Cameroon. It is a city community consisting of seven municipalities. The estimated population was 4.1 million residents in 2020. Its geographic coordinates are 3° 52' 12" North and 11° 31' 12" East, at an elevation of 750 m above sea level. The city of Yaoundé features a tropical savanna climate, although tempered by altitude and a relative humidity exceeding 80%. The average monthly temperature ranges from 23°C (August) to 26°C (February). It is characterized by several months of heavy rainfall, with a slight lull in July-August, and a dry season that lasts from the end of November to February [24]. Owing to the dense population, tight living conditions, and lack of proper waste management and treatment facilities, Yaoundé ranks among the cities in Cameroon where malaria and typhoid fever continue to be significant health problems.

### Study design and target population

From October 2023 to May 2024, this cross-sectional study followed by laboratory investigations targeted all febrile patients who came for consultation at the CMA of Obili or at the CHD of Mvog-Betsi, and who presented at least one of the following clinical symptoms: fever, headache, abdominal pain, muscle aches, joint pain, nausea, vomiting, and chills. Overall, the inclusion criteria were: patients presenting at least one of the clinical signs of malaria or typhoid fever, patients who voluntarily, freely, and without coercion agreed to participate in the study by signing the informed consent form and the guardians of children who provided their consent. Patients arriving in a critical condition, patients suffering from hepatitis or with a recent history of liver disease were excluded from the study.

### Sampling technique and sample size estimation

Participants were recruited through convenience random sampling. After approaching a potential participant, the goals of the study, the procedures, as well as the potential risks and benefits were clearly explained to them; and only participants who gave their consent by signing the informed consent form were enrolled in the study (Additional information S1 File). The estimated sample size for this study was calculated using the Lorentz formula (1).

$$N = \frac{Z^2 \times p(1-p)}{d^2}$$

(1)

Where: $N$ is the calculated sample size; $Z = 1.96$ is the typical value of the degree of confidence at 95%; $p = 77.01\%$ is the prevalence of malaria among febrile patients [25]; $d = 5\%$ is the margin error permitted.

After calculation, $N \approx 322$ participants. For a better representation of the population, the minimum sample size was rounded to 350 participants.

## Ethical considerations

This study was conducted in accordance with the guidelines of the Declaration of Helsinki. The study protocol was reviewed and approved by the Joint Institutional Review Board for Animal & Human Bioethics of the University of Yaoundé I, which subsequently issued an ethical clearance (reference number: BTC-JIRB2023–072). In addition, administrative authorizations were granted respectively by the directors of the CMA of Obili and the CHD of Mvog-Betsi. The information collected during the interview was not disclosed, and the participant's names were replaced with codes during the data collection.

## Data collection procedure

**Determination of patients' attitudes towards adherence to preventive measures and management against malaria and typhoid fever.**  The outlook of febrile patients seeking consultation at the CMA or CHD about the routine implementation of barrier measures and the treatment of malaria and typhoid fever was determined through a structured questionnaire (Additional information S1 File) given in person. The survey was organized into four main sections to gather information as follows:

- The socio-demographic characteristics of the patient encompass gender, age, marital status, level of educational, profession, housing type, toilet facilities, and the sources of drinking and running water.

- The clinical symptoms experienced by the patient include fever, headaches, chills, muscles aches, join pain, vomiting, nausea, abdominal pain, fatigue, diarrhea, etc.

- The respect of the preventive measures and factors that facilitate the transmission of the disease, including the presence of bushes or crops, stagnant water, and garbage deposits near homes; the presence or absence of a ceiling in the house, the availability of mosquito nets on windows, the regularity of the use of insecticides or mosquito nets, the washing of fruits and vegetables before consumption, hand hygiene before meals, and the intake of preventive medication or vaccines.

- The patient's approach to proper disease management and treatment, which involved the duration before consulting a professional from the emergence of initial symptoms, the care setting (hospital care, self-treatment, herbal remedies), and the type of medication administered.

At the end of the interview, 5 mL of blood were collected via venipuncture into a tube containing the anticoagulant (Ethylene Diamine Tetra-Acetic acid tube) and another tube free of anticoagulant (dry tube). The blood in the EDTA tube was used for the microscopic diagnosis of malaria, while the blood in the dry tube was centrifuged (3000 × g, 5 min, 4°C). The obtained serum was used for the diagnosis of typhoid fever and the measurement of some biochemical markers of liver damage.

**Determination of the prevalence of malaria and typhoid fever among febrile patients.**  The detection of the *Plasmodium* parasite responsible for malaria in febrile patients was done by microscopy using Giemsa staining as reported elsewhere [6]. Every positive slide was quantified by tallying the number of parasites against 200 white blood cells, and the parasite density or parasitemia (number of parasites/μL of blood) was estimated by assuming a leucocyte count of 8000/μL as previously described reported [26]. The diagnosis of typhoid fever was performed through the Widal serological test, which relies on the principle of mixing a suspension of somatic antigens from the cell wall (O antigen) and flagellar antigens (H antigen) of Salmonella bacteria with the patient's serum, allowing the visualization of agglutination if the specific antibodies are present. The Widal assay kit (Ref N° IS-SAW.078, RECKON DIAGNOSTICS P. LTD. Gorwa Vadodara, India) was used to conduct the test in two stages: first, a qualitative test on a glass slide to identify specific antibodies against the *Salmonella* genus, followed by a semi-quantitative tube test to confirm the result observed on the glass slide. The procedure was

performed according to the manufacturer's instructions. Finally, the prevalence of malaria or typhoid fever, and the parasitemia (number of parasites/µL of blood) for every malaria-positive patient were determined using equations (2) and (3), respectively.

$$\textit{Prevalence of malaria or typhoid fever } (\%) = \frac{\text{Number of positif patients}}{\text{Total number of tested febrile patients}} \times 100 \tag{2}$$

$$\textit{Parasitemia } (\textit{parasites}/\mu L) = \frac{\text{Total number of parasite/200 white blood cells}}{\text{Total number of white blood cells/field}} \times 8000 \tag{3}$$

For more sensitivity, a parasite density lower than 500, between 500 and 2500, or higher than 2500 parasites/µL was considered low, medium, or high, respectively. This classification was adapted from the study of Sumbele et al. [27].

**Evaluation of some biochemical markers of liver damage in infected patients.** The evaluation of biochemical markers of liver injury in malaria, typhoid or malaria-typhoid co-infected febrile patients was carried out by measuring the serum activity of some liver enzymes, including alanine aminotransferase (ALT), aspartate aminotransferase (AST), and alkaline phosphatase (ALP) using commercial assay kits (Catalog N° REF_80227, Catalog N° REF_80225, and Catalog N° REF_80014 respectively for ALT, AST and ALP) purchased from BIOLABO, Les Hautes Rives, Maizy, France. The assays were performed in accordance with the manufacturer's instructions using a semi-automatic spectrophotometer.

**Estimation of the level of alteration of serum liver enzyme activities.** The level of alteration of the serum liver enzyme activities in feverish patients suffering from malaria and/or typhoid fever was estimated by considering the normal range of values (reference values) for each parameter evaluated. In our laboratory, these reference values in humans are: from 10 to 42 UI/L, from 8 to 39 UI/L, and from 40 to 129 UI/L respectively for ALT, AST, and ALP. Accordingly, any value found: within its normal range, less than 2 times the upper limit of the normal range of values (ULN), between 2 and 5 times the ULN, or higher than 5 times the ULN, the level of alteration was qualified as "normal", "mild", "moderate", and "severe", respectively.

**Estimation of the type of liver damages.** The type of liver damage caused by the impact of malaria and/or typhoid fever in febrile patients was estimated using the "RUCAM" method (Roussel Uclaf Causality Assessment Method) [28]. The RUCAM method consists of determining whether the hepatic injury is "hepatocellular" (cytolysis), "cholestatic", or "mixed". It is based on the profile of serum liver enzymes activities, particularly ALT and ALP, and the type of liver damage is defined based on the "R Ratio," calculated using formula (4).

$$R\ Ratio = \frac{(ALT_{value} \div ALT_{ULN})}{(ALP_{value} \div ALP_{ULN})} \tag{4}$$

Where: $ALT_{value}$ and $ALP_{value}$ are the respective values of ALT and ALP obtained after analysis, while $ALT_{ULN}$ and $ALP_{ULN}$ are the upper limit of the normal range of values for ALAT and ALP, respectively.

Thus, R ratios > 5, < 2, or between 2 and 5 respectively define a hepatocellular, a cholestatic, or a mixed pattern of hepatocellular/cholestasis. However, if the ALT value is more than twice the ULN and the ALP value is normal, the hepatic damage is considered as hepatocellular, and it is not necessary to calculate the R ratio. Similarly, if the ALP value obtained after the analysis is more than twice the ULN and the ALT value is normal, the damage is considered cholestatic, and the R ratio need not to be determined.

## Data management and statistical analysis

After checking that all sections of the questionnaire had been completed, the data collected and the results of the laboratory analyses for each participant were saved in Excel 2016 (Microsoft Corporation, USA) (Additional information S3 File), and then exported to the statistical analysis software SPSS (Statistical Package for Social Sciences) version 25.0

(SPSS Inc., USA) or GraphPad Prism version 8.0.2. Descriptive statistics were performed using SPSS software. Qualitative variables were presented as frequency and percentage (%). Quantitative variables were first tested for normality using the Kolmogorov-Smirnov test. Variables that followed a normal distribution and those that did not pass the test were expressed as mean ± standard deviation or median-interquartile range respectively. Comparison of median values between two groups was done by the non-parametric Mann Whitney $U$ test. The Chi-square test or Fisher's exact with the Relative Risk (RR) were calculated from the contingency tables to determine the risk factors associated with malaria, typhoid or malaria-typhoid co-infections, the alteration of serum activities of liver enzymes, and the type of hepatic damage. The significance threshold was declared at $P < 0.05$.

## Results

### Distribution of febrile patients enrolled in the study according to their socio-demographic characteristics

The distribution of febrile patients who attended consultations at the CMA of Obili and CHD of Mvog-Betsi and participated in this study based on their socio-demographic characteristics is presented in the Additional information S2 File, Table S1). Out of the 350 febrile patients enrolled in the study, the gender ratio was 1.97 in favor of women, who constituted 66.3% (232/350) of the participants. The median age was 28 years, with interquartile ranges of 25% and 75% being 20 and 42.25 years, respectively. The most represented age group was between}20–40] years (48.9%; 171/350), while 19 patients were under 5 years of age. Regarding marital status and education level, 57.7% (202/350) were single, while 56.9% (199/350) were university attendants. Less than one-fifth (13.4%; 47/350) of the patients were civil servants, and a large majority (86.6%; 310/350) lived in cement houses. Nearly one-third (32.6%; 114/350) of the participants used pit latrine toilet, and the use of borehole water represented 47.1% (165/350) and 60.3% (211/350) as sources of running water and drinking water, respectively.

### Distribution of the study population according to the clinical symptoms experienced

The clinical symptoms experienced by the participants of this study were recorded and presented in the Additional information S2 File, Table S2. The majority (83.4%; 292/350) of participants experienced fever, while less than half (47.7%; 167/350) had headaches. Regarding chills, asthenia, and muscle aches, the proportions were 32.9% (115/350), 49.7% (174/350), and 32.6% (114/350), respectively. The patients who experienced abdominal pain, diarrhea, and vomiting, were 33.1% (116/350), 13.1% (46/350), and 22.0% (77/350), respectively.

### Distribution of the participants based on their attitude towards the practice of preventive measures against malaria and typhoid fever

It appears that 62.3% (218/350) and 14.0% (49/350) were regular users of mosquito nets and insecticides, respectively. Additionally, 28.3% (99/350), 17.1% (60/350), 15.4% (54/350), 41.7% (113/350), and 18.0% (63/350) of patients reported staying out late at night, having standing water, crops, bushes, and garbage dumps near their homes, respectively. Only 7.1% (25/350) of patients declared they took prophylaxis against malaria. The majority of participants were in favor of the practice of handwashing before meals (88.6%; 310/350) and washing fruits and vegetables before consumption (92.0%; 322/350). Conversely, the majority of patients (77.7%; 272/350) stated not to purify water before drinking (Additional information S2 File, Table S3).

### Distribution of the feverish patients based on their attitude towards management and treatment of malaria and typhoid fever

From the survey, 48.3% (169/350) and 7.4% (26/350) of participants attended the hospital 3 and 7 days after the onset of the first symptoms, respectively. The proportions of participants who referred to a health center for the treatment of

malaria and typhoid fever were 60.9% (213/350) and 75.7% (265/350), respectively. Self-medication was more practiced for the treatment of malaria (32.0%; 112/350) compared to typhoid fever (4.0%; 14/350). In addition, participants turned more to traditional medicine to treat typhoid fever (20.3%; 71/350) compared to malaria (7.1%; 25/350). The Artemisinin-Based Combination Therapy (ACT) (73.4%; 257/350) and antibiotics: Ceftriaxone (49.7%; 174/350) and Ciprofloxacin (30.0%; 105/350) were the main medications used for the treatment of malaria and typhoid fever, respectively (Additional information S2 File, Table S4).

### Seroprevalence of malaria, typhoid fever, and malaria-typhoid co-infection among febrile patients

Out of the 350 febrile patients enrolled, 179 participants tested positive for at least one infection, resulting in an overall incidence of 51.14% (Fig 1A). Distinguishing between each infection, 129/350 (36.86%) participants were tested positive for malaria (Fig 1B), while 106/350 (30.29%) patients were tested positive for typhoid fever (Fig 1C). Furthermore, the Venn diagram (Fig 1D) showing the prevalence of mono-infections and co-infections reveals that among the 179

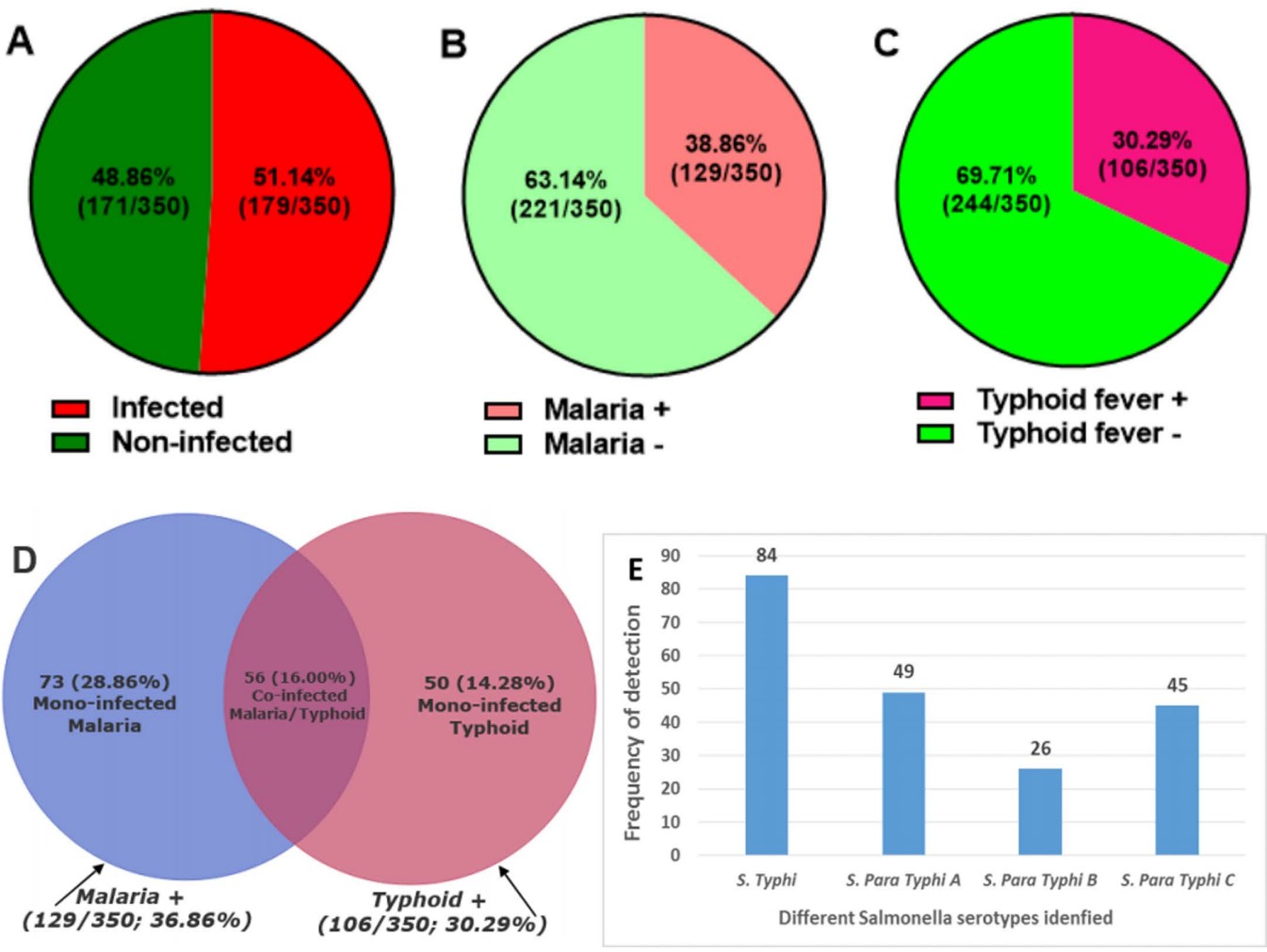

**Fig 1. Prevalence of malaria and typhoid fever in the study population.** *(A): Overall prevalence of at least one infection; (B): Prevalence of malaria; (C): Prevalence of typhoid fever; (D): Venn diagram showing the rate of co-infection between malaria/typhoid fever; (E): Frequency of detection of different serotypes of Salmonella typhi in patients suffering from typhoid fever.*

patients with at least one infection, malaria mono-infection, typhoid fever mono-infection, and malaria/typhoid co-infection accounted for 20.86% (73/350), 14.28% (50/350), and 16.00% (56/350) respectively. Also, Fig 1E highlights the different Salmonella serotypes of identified from typhoid-infected patients. The predominant species was *Salmonella typhi* (84 cases), followed by *Salmonella para-typhi A* (49 cases), *Salmonella para-typhi C* (45 cases), and *Salmonella para-typhi B* (26 cases).

### Effect of socio-demographic characteristics on malaria and typhoid fever infection rate among febrile patients

Table 1 summarizes the statistical association between malaria and the various socio-demographic of febrile patients consulting at the CMA of Obili and the CHD of Mvog-Betsi. For both malaria and typhoid fever, men (58/118 and 44/118) were more affected than women (71/232 and 62/232), with significant relative risks (RR) of [RR = 1.61, P = 0.001] and [RR = 1.40; P = 0.049], respectively. Aged groups:}0–5],}5–20], and}20–40] were significantly (P < 0.05) three times more at risk (RR > 3) of being affected by malaria compared to patients above 60 years. No significant association (RR ≈ 1; P > 0.05) was found between age group, marital status, level of education, profession, and typhoid fever. However, being single (92/202; RR = 3.41; P = 0.015) and having attained university (83/199; RR = 1.55; P = 0.011) represented a significant risk for malaria infection. Additionally, households with pit latrine (51/114; RR = 1.92; P < 0.0001) and the use of tap water (39/78; RR = 2.38; P < 0.0001) as a source of running water were significantly (P < 0.05) associated with typhoid fever.

### Effect of clinical symptoms on the rates of malaria and typhoid fever infection in febrile patients

The association between the clinical symptoms experienced by the febrile patients and malaria or typhoid fever infection is presented in Table 2. A significant association (RR > 1; P < 0.05) was observed between the presence of fever (116/292; RR = 1.77; P = 0.016), headaches (95/167; RR = 3.06; P < 0.0001), chills (73/115; RR = 2.66; P < 0.0001), muscle aches (73/114; RR = 2.69; P < 0.0001), vomiting (36/77; RR = 1.37; P = 0.045), and the detection of malaria parasite. Similarly, a significant association (RR > 1; P < 0.05) was observed between patients who experienced abdominal pain (76/116; RR = 4.27; P < 0.0001), asthenia (75/174; RR = 2.44; P < 0.0001), vomiting (40/77; RR = 2.14; P < 0.0001), and the detection of *Salmonella* serotypes.

### Influence of the attitudes of febrile patients towards adherence to preventive measures on malaria and typhoid fever infections rate

Table 3 summarizes the association between patients' attitudes regarding adherence to preventive measures and the rate of malaria infection. Risky behaviors such as staying outside late at night (45/99; RR = 1.35; P = 0.048), irregular use of mosquito nets (35/55; RR = 2.75; P < 0.0001), or non-use of mosquito nets (48/77; RR = 2.83; P < 0.0001) were significantly (RR > 1; P < 0.05) associated with malaria infection. Likewise, environmental factors such as the presence of standing water (38/60; RR = 2.01; P < 0.0001) and bushes (57/113; RR = 1.66; P = 0.0004) near residential homes represented significant risk factors (RR > 1; P < 0.05) for contracting malaria. Regarding typhoid fever, as indicated in Table 4, only the presence of garbage dumps near residential homes (36/63; RR = 2.34; P < 0.0001) constituted a significant risk (RR > 1; P < 0.05) for being infected by *Salmonella typhi* serotypes. Other factors such as the purification of water prior drinking, hand washing before meals, and fruits of vegetables washing before consumption were not significantly (RR < 1; P > 0.05) associated with typhoid fever.

### Variation of biochemical markers of liver damage in malaria, typhoid, and malaria-typhoid co-infected patients

**Influence of infection status and parasitemia on the liver damage parameters in febrile patients.** Fig 2A, Fig 2B, and Fig 2C present the influence of the infection status on serum levels of ALT, AST, and ALP respectively. The serum levels of these biochemical markers of liver injury were significantly (P < 0.05) higher in co-infected patients compared to

**Table 1. Association between malaria or typhoid fever infection and the socio-demographic characteristics.**

| Variables | Categories | Malaria | | | | | Typhoid fever | | | | |
|---|---|---|---|---|---|---|---|---|---|---|---|
| | | Positive n (%) | Negative n (%) | RR | [95% CI] | P-value | Positive n (%) | Negative n (%) | RR | [95% CI] | P-value |
| **Gender** | Male | 58 (16.6) | 60 (17.1) | **1.61** | [1.2 – 2.1] | **0.001\*** | 44 (12.6) | 74 (21.1) | **1.40** | [1.01 - 1.90] | **0.049\*** |
| | Female | 71 (20.3) | 161 (46.0) | 1.00 | / | / | 62 (17.7) | 170 (48.6) | 1.00 | / | / |
| | **Total** | **129 (36.9)** | **221 (63.1)** | / | / | / | **106 (30.3)** | **244 (69.7)** | / | / | / |
| **Age group** | ]0–5] | 9 (2.6) | 10 (2.9) | **3.88** | [1.5 – 9.8] | **0.006\*** | 5 (1.4) | 14 (4.0) | 1.32 | [0.5 – 3.1] | 0.314 |
| | ]5–20] | 29 (8.4) | 39 (11.1) | **3.49** | [1.6 – 8.3] | **0.001\*** | 21 (6.0) | 47 (13.4) | 1.57 | [0.8 – 3.0] | 0.207 |
| | ]20–40] | 75 (21.4) | 96 (27.4) | **3.59** | [1.7 – 8.4] | **0.000\*** | 61 (17.4) | 110 (31.4) | **1.82** | [1.0 – 3.3] | **0.039\*** |
| | ]40–60] | 11 (3.1) | 40 (11.4) | 1.76 | [0.7 – 4.6] | 0.279 | 10 (2.9) | 41 (11.7) | 1.00 | / | / |
| | >60 | 5 (1.4) | 36 (10.3) | 1.00 | / | / | 9 (2.6) | 32 (9.1) | 1.12 | [0.5 – 2.4] | 0.801 |
| | **Total** | **129 (36.9)** | **221 (63.1)** | / | / | / | **106 (30.3)** | **244 (69.7)** | / | / | / |
| **Marital status** | Single | 92 (26.3) | 110 (31.4) | **3.41** | [1.2 – 12.2] | **0.015\*** | 74 (21.1) | 128 (36.6) | 2.19 | [0.6 – 12.2] | 0.4 |
| | Married | 32 (9.1) | 95 (27.1) | 1.89 | [0.6 – 6.9] | 0.522 | 28 (8.0) | 99 (28.3) | 1.3 | [0.3 – 7.4] | >0.999 |
| | Divorced | 3 (0.9) | 3 (0.9) | 3.75 | [0.9 – 14.4] | 0.144 | 1 (0.3) | 5 (1.4) | 1.00 | / | / |
| | Widow(er) | 2 (0.6) | 13 (3.7) | 1.00 | / | / | 3 (0.9) | 12 (3.4) | 1.20 | [0.2 – 7.8] | >0.999 |
| | **Total** | **129 (36.9)** | **221 (63.1)** | / | / | / | **106 (30.3)** | **244 (69.7)** | / | / | / |
| **Level of education** | Not-schooled | 5 (1.4) | 4 (1.1) | 2.06 | [0.6 – 6.9] | 0.119 | 3 (0.9) | 6 (1.7) | 1.33 | [0.4 – 2.9] | 0.692 |
| | Primary | 13 (3.7) | 26 (7.4) | 1.23 | [0.6 – 6.9] | 0.534 | 12 (3.4) | 27 (7.7) | 1.23 | [0.6 – 2.1] | 0.526 |
| | Secondary | 28 (8.0) | 76 (21.7) | 1.00 | / | / | 26 (7.4) | 78 (22.3) | 1.00 | / | / |
| | University | 83 (23.7) | 115 (32.9) | **1.55** | [1.1 – 2.2] | **0.011\*** | 65 (18.6) | 133 (38.0) | 1.31 | [0.9 – 1.9] | 0.187 |
| | **Total** | **129 (36.9)** | **221 (63.1)** | / | / | / | **106 (30.3)** | **244 (69.7)** | / | / | / |
| **Profession** | Employee | 24 (6.9) | 46 (13.1) | 1.34 | [0.7 – 2.4] | 0.414 | 11 (3.1) | 36 (10.3) | 1.00 | / | / |
| | Self-employee | 93 (26.6) | 140 (40.0) | 1.56 | [0.9 – 2.7] | 0.070 | 25 (7.1) | 45 (12.9) | 1.52 | [0.8 – 2.8] | 0.220 |
| | Civil servant | 12 (3.4) | 35 (10.0) | 1.00 | / | / | 70 (20.0) | 163 (46.6) | 1.28 | [0.9 – 2.2] | 0.480 |
| | **Total** | **129 (36.9)** | **221 (63.1)** | / | / | / | **106 (30.3)** | **244 (69.7)** | / | / | / |
| **Type of house** | Cement | 118 (33.7) | 192 (54.8) | 1.71 | [0.6 – 6.0] | 0.491 | 98 (28.0) | 212 (60.6) | 1.96 | [0.9 – 4.5] | 0.099 |
| | Mud | 9 (2.6) | 22 (6.3) | 1.30 | [0.4 – 4.9] | >0.999 | 5 (1.4) | 26 (7.4) | 1.00 | / | / |
| | Wood | 2 (0.6) | 7 (2.00) | 1.00 | / | / | 3 (0.9) | 6 (1.7) | 2.06 | [0.6 – 6.2] | 0.347 |
| | **Total** | **129 (36.9)** | **221 (63.1)** | / | / | / | **106 (30.3)** | **244 (69.7)** | / | / | / |
| **Type of toilet** | Pit latrine | 50 (14.3) | 64 (18.3) | 1.31 | [0.9 – 1.7] | 0.075 | 51 (14.6) | 63 (18.00) | **1.92** | [1.4 – 2.6] | **<0.0001\*** |
| | Water system | 79 (22.6) | 157 (44.8) | 1.00 | / | / | 55 (15.7) | 181 (51.7) | 1.00 | / | / |
| | **Total** | **129 (36.9)** | **221 (63.1)** | / | / | / | **106 (30.3)** | **244 (69.7)** | / | / | / |
| **Source of running water** | Borehole | 52 (14.8) | 113 (32.3) | 1.02 | [0.5 – 1.9] | >0.999 | 42 (12.00) | 123 (35.2) | 1.21 | [0.7 – 2.0] | 0.525 |
| | Well | 27 (7.7) | 54 (15.4) | 1.08 | [0.5 – 2.1] | >0.999 | 17 (4.8) | 64 (18.3) | 1.00 | / | / |
| | Tap water | 42 (12.0) | 36 (10.3) | **1.75** | [1.0 – 3.3] | **0.045\*** | 39 (11.1) | 39 (11.1) | **2.38** | [1.5 – 3.8] | **<0.0001\*** |
| | Spring | 8 (2.3) | 18 (5.1) | 1.00 | / | / | 8 (2.3) | 18 (5.1) | 1.4 | [0.7 – 2.8] | 0.301 |
| | **Total** | **129 (36.9)** | **221 (63.1)** | / | / | / | **106 (30.3)** | **244 (69.7)** | / | / | / |
| **Source of drinking water** | Mineral | 20 (5.7) | 34 (9.7) | 1.33 | [0.6 – 3.1] | 0.574 | 38 (10.9) | 29 (8.3) | **2.65** | [1.8 – 3.7] | **<0.0001\*** |
| | Borehole | 73 (20.9) | 138 (39.4) | 1.24 | [0.6 – 2.8] | 0.616 | 45 (12.8) | 166 (47.4) | 1.00 | / | / |
| | Tap water | 31 (8.9) | 36 (10.3) | 1.66 | [0.8 – 3.8] | 0.188 | 16 (4.6) | 38 (10.9) | 1.38 | [0.8 – 2.2] | 0.207 |
| | Spring | 5 (1.4) | 13 (3.7) | 1.00 | / | / | 7 (2.00) | 11 (3.1) | 1.82 | [0.9 – 3.1] | 0.137 |
| | **Total** | **129 (36.9)** | **221 (63.1)** | / | / | / | **106 (30.3)** | **244 (69.7)** | / | / | / |

*The statistical association was performed using Fisher's Exact Test. P values <0.05 were considered significantly different. RR: Relative Risk; CI: 95% Confidence Interval; n: sample size of the category. All categories where RR = 1.00 were considered as reference during the analysis.*

**Table 2. Association between malaria or typhoid fever infection and the clinical symptoms experienced by febrile patients.**

| Clinical symptoms | Categories | Malaria | | | | | Typhoid fever | | | | |
|---|---|---|---|---|---|---|---|---|---|---|---|
| | | Positive n (%) | Negative n (%) | RR | [95% CI] | P-value | Positive n (%) | Negative n (%) | RR | [95% CI] | P-value |
| Fever | Yes | 116 (33.2) | 176 (50.3) | **1.77** | [1.1 – 2.9] | **0.016*** | 92 (26.3) | 200 (57.1) | 1.30 | [0.8 – 2.1] | 0.347 |
| | No | 13 (3.7) | 45 (12.8) | 1.00 | / | / | 14 (4.0) | 44 (12.6) | 1.00 | / | / |
| | Total | **129 (36.9)** | **221 (63.1)** | / | / | / | **106 (30.3)** | **244 (69.7)** | / | / | / |
| Headaches | Yes | 95 (27.2) | 72 (20.6) | **3.06** | [2.2 – 4.2] | **<0.0001*** | 55 (15.7) | 112 (32.0) | 1.18 | [0.8 – 1.6] | 0.351 |
| | No | 34 (9.7) | 149 (42.5) | 1.00 | / | / | 51 (14.6) | 132 (37.7) | 1.00 | / | / |
| | Total | **129 (36.9)** | **221 (63.1)** | / | / | / | **106 (30.3)** | **244 (69.7)** | / | / | / |
| Chills | Yes | 73 (20.9) | 42 (12.0) | **2.66** | [2.0 – 3.4] | **<0.0001*** | 39 (11.2) | 76 (21.7) | 1.18 | [0.8 – 1.6] | 0.323 |
| | No | 56 (16.0) | 179 (51.1) | 1.00 | / | / | 67 (19.1) | 168 (48.0) | 1.00 | / | / |
| | Total | **129 (36.9)** | **221 (63.1)** | / | / | / | **106 (30.3)** | **244 (69.7)** | / | / | / |
| Muscle aches | Yes | 73 (20.9) | 41 (11.7) | **2.69** | [2.0 – 3.5] | **<0.0001*** | 33 (9.4) | 81 (23.1) | 1.00 | / | / |
| | No | 56 (16.0) | 180 (51.4) | 1.00 | / | / | 73 (20.9) | 163 (46.6) | 1.06 | [0.7 – 1.5] | 0.804 |
| | Total | **129 (36.9)** | **221 (63.1)** | / | / | / | **106 (30.3)** | **244 (69.7)** | / | / | / |
| Abdominal pain | Yes | 38 (10.9) | 78 (22.3) | 1.00 | / | / | 72 (20.6) | 44 (12.6) | **4.27** | [3.0 – 6.0] | **<0.0001*** |
| | No | 91 (26.0) | 143 (40.8) | 1.18 | [0.6 – 1.1] | 0.290 | 34 (9.7) | 200 (57.1) | 1.00 | / | / |
| | Total | **129 (36.9)** | **221 (63.1)** | / | / | / | **106 (30.3)** | **244 (69.7)** | / | / | / |
| Asthenia | Yes | 62 (17.7) | 112 (32.0) | 1.00 | / | / | 75 (21.4) | 99 (28.3) | **2.44** | [1.7 – 3.5] | **<0.0001*** |
| | No | 67 (19.2) | 109 (31.1) | 1.06 | [0.8 – 1.4] | 0.658 | 31 (8.9) | 145 (41.4) | 1.00 | / | / |
| | Total | **129 (36.9)** | **221 (63.1)** | / | / | / | **106 (30.3)** | **244 (69.7)** | / | / | / |
| Diarrhea | Yes | 12 (3.5) | 34 (9.7) | 1.00 | / | / | 19 (5.4) | 27 (7.7) | 1.44 | [0.9 – 2.0] | 0.087 |
| | No | 117 (33.4) | 187 (53.4) | 1.47 | [0.9 – 2.5] | 0.139 | 87 (24.9) | 217 (62.0) | 1.00 | / | / |
| | Total | **129 (36.9)** | **221 (63.1)** | / | / | / | **106 (30.3)** | **244 (69.7)** | / | / | / |
| Vomiting | Yes | 36 (10.3) | 41 (11.7) | **1.37** | [1.0 – 1.8] | **0.045*** | 40 (11.4) | 37 (10.6) | **2.14** | [1.5 – 2.8] | **<0.0001*** |
| | No | 93 (26.6) | 180 (51.4) | 1.00 | / | / | 66 (18.9) | 207 (59.1) | 1.00 | / | / |
| | Total | **129 (36.9)** | **221 (63.1)** | / | / | / | **106 (30.3)** | **244 (69.7)** | / | / | / |

*The statistical association was performed using Fisher's Exact Test. P values <0.05 were considered significantly different. RR: Relative Risk; CI: 95% Confidence Interval; n: sample size of the category. All categories where RR = 1.00 were considered as reference during the analysis.*

those with malaria or typhoid mono-infection. Similarly, Fig 2D, Fig 2E, and Fig 2F depict the effect of parasitemia on the serum activities of ALT, AST, and ALP in patients suffering from malaria. No significant (P > 0.05) variation in the levels of these liver enzymes was observed between patients with a parasitemia lower than 500 parasites/µL and those with parasitemia between 500 and 2500 parasites/µL. However, the serum activities of these liver enzymes were significantly (P < 0.05) higher in patients with parasitemia above 2500 parasites/µL, when compared to the previous two groups. Nonetheless, regardless of the patient's group considered, the median values of each of these parameters were at least twice as high as the upper limit of the normal range of their respective reference values.

**Influence of the time taken before consultation and the place of treatment on the variation of biochemical markers of liver injury in febrile patients.** The effects of the time taken before consultation since the onset of the first symptoms on serum levels of ALT, AST, and ALP are illustrated in Fig 3A, Fig 3B, and Fig 3C respectively. There was no significant change (P > 0.05) in the levels of liver enzymes activities between patients who spent less than three days and those whose consultation time varied between three and seven days since the appearance of the first symptoms. However, the serum activity of these liver enzymes was significantly (P < 0.05) higher in infected patients whose time taken before consultation exceeded seven days, compared to the previous two groups.

**Table 3. Association between malaria infection rate and patients' attitudes towards compliance with preventive measures.**

| Preventive measures | Categories | Malaria | | RR | [95% CI] | P-value |
|---|---|---|---|---|---|---|
| | | Positive n (%) | Negative n (%) | | | |
| **Stay out late at night** | Yes | 45 (12.9) | 54 (15.4) | **1.35** | [1.0 – 1.7] | **0.048*** |
| | No | 84 (24.0) | 167 (47.7) | 1.00 | / | / |
| | **Total** | **129 (36.9)** | **221 (63.1)** | / | / | / |
| **Standing water near the house** | Yes | 38 (10.9) | 22 (6.3) | **2.01** | [1.5 – 2.5] | **<0.0001*** |
| | No | 91 (26.0) | 199 (56.8) | 1.00 | / | / |
| | **Total** | **129 (36.9)** | **221 (63.1)** | / | / | / |
| **Crops near the house** | Yes | 22 (6.3) | 32 (9.1) | 1.12 | [0.7 – 1.5] | 0.541 |
| | No | 107 (30.6) | 189 (54.0) | 1.00 | / | / |
| | **Total** | **129 (36.9)** | **221 (63.1)** | / | / | / |
| **Bushes near the house** | Yes | 57 (16.3) | 56 (16.0) | **1.66** | [1.2 – 2.1] | **0.0004*** |
| | No | 72 (20.6) | 165 (47.1) | 1.00 | / | / |
| | **Total** | **129 (36.9)** | **221 (63.1)** | / | / | / |
| **Windows with nets** | Yes | 53 (15.2) | 93 (26.5) | 1.00 | / | / |
| | No | 76 (21.7) | 128 (36.6) | 1.02 | [0.7 – 1.3] | 0.910 |
| | **Total** | **129 (36.9)** | **221 (63.1)** | / | / | / |
| **Use of mosquito nets** | Regular | 48 (13.7) | 170 (48.5) | 1.00 | / | / |
| | Irregular | 33 (9.4) | 22 (6.3) | **2.75** | [1.9 – 3.7] | **<0.0001*** |
| | Don't use | 48 (13.7) | 29 (8.3) | **2.83** | [2.0 – 3.8] | **<0.0001*** |
| | **Total** | **129 (36.9)** | **221 (63.1)** | / | / | / |
| **Use of insecticide mosquito spray** | Regular | 24 (6.86) | 25 (7.1) | 1.00 | / | / |
| | Irregular | 15 (4.28) | 31 (8.9) | 0.66 | [0.3 – 1.0] | 0.144 |
| | Don't use | 90 (25.71) | 165 (47.1) | 0.72 | [0.5 – 1.0] | 0.077 |
| | **Total** | **129 (36.9)** | **221 (63.1)** | / | / | / |
| **House with ceiling** | Yes | 120 (34.3) | 199 (56.8) | 1.00 | / | / |
| | No | 9 (2.6) | 22 (6.3) | 0.77 | [0.4 – 1.2] | 0.436 |
| | **Total** | **129 (36.9)** | **221 (63.1)** | / | / | / |
| **Malaria prophylaxis** | Yes | 7 (2.0) | 18 (5.1) | 1.00 | / | / |
| | No | 122 (34.9) | 203 (58.0) | 1.3 | [0.7 – 2.6] | 0.395 |
| | **Total** | **129 (36.9)** | **221 (63.1)** | / | / | / |

*The statistical association was performed using Fisher's Exact Test. P values <0.05 were considered significantly different. RR: Relative Risk; CI: 95% Confidence Interval; n: sample size of the category. All categories where RR = 1.00 were considered as reference during the analysis.*

Furthermore, Fig 3D, 3E, and 3F depict the influence of the place of treatment on the variation of serum activities of ALT, AST, and ALP, respectively. Regardless of the location where infected patients seek health care, no significant (P > 0.05) variation in the serum levels of these liver enzymes was observed between patients who seek treatment from a health center or from a traditional practitioner/herbalism and those who engaged in self-medication. Nevertheless, the median values of each of these parameters were more than twice the upper limit of the normal range of their respective reference values.

### Risk factors associated with the alteration of the serum activities of hepatic enzymes

**Association between the infection status and the levels of alteration of hepatic enzymes.** Table 5 presents the associations between the infection status (malaria mono-infected, typhoid fever mono-infected, or malaria/

**Table 4. Association between typhoid fever infection rate and patients' attitudes towards compliance with preventive measures.**

| Preventive measures | Categories | Typhoid fever | | | | |
| --- | --- | --- | --- | --- | --- | --- |
| | | Positive n (%) | Negative n (%) | RR | [95% CI] | P-value |
| purify water before drinking | Yes | 30 (8.6) | 48 (13.7) | 1.00 | / | / |
| | No | 76 (21.7) | 196 (56.0) | 0.72 | [0.5 – 1.0] | 0.093 |
| | Total | **106 (30.3)** | **244 (69.7)** | / | / | / |
| washing fruits and vegetables before consumption | Yes | 98 (28.0) | 224 (64.0) | 1.00 | / | / |
| | No | 8 (2.3) | 20 (5.7) | 0.938 | [0.4 – 1.5] | >0.999 |
| | Total | **106 (30.3)** | **244 (69.7)** | / | / | / |
| hand washing before meals | Yes | 96 (27.4) | 214 (61.1) | 1.00 | | |
| | No | 10 (2.9) | 30 (8.6) | 0.80 | [0.4 – 1.5] | 0.583 |
| | Total | 106 (30.3) | 244 (69.7) | / | / | / |
| garbage dumps near the house | Yes | 36 (10.3) | 27 (7.7) | **2.34** | [1.7 – 3.1] | **<0.0001*** |
| | No | 70 (20.0) | 217 (62.0) | 1.00 | / | / |
| | Total | **106 (30.3)** | **244 (69.7)** | / | / | / |

*The statistical association was performed using Fisher's Exact Test. P values <0.05 were considered significantly different. RR: Relative Risk; CI: 95% Confidence Interval; n: sample size of the category. All categories where RR = 1.00 were considered as reference during the analysis.*

typhoid co-infected) and the different levels of alterations in liver enzyme activity (normal, mild, moderate, and severe). The infection status was not significantly associated with the level of alteration of ALT activity (Chi-square = 2.13; P = 0.343). However, a significant association was found between the infection status and the levels of alteration of AST activity (Chi-square = 8.44; P = 0.014) and ALP activity (Chi-square = 13.35; P = 0.001), respectively.

**Association between the parasitemia and the levels of alteration of hepatic enzymes.** The statistics on the associations between parasitemia (<500;}500–2500];>2500 parasites/μL of blood) in malaria-infected patients and the different levels of alteration in liver enzyme activity are summarized in Table 6. A significant association was noted between parasitemia and the levels of alteration in ALT activity (Chi-squared = 9.36; P = 0.009) and AST activity (Chi-squared = 13.75; P = 0.001), respectively. No significant association (Chi-squared = 1.48; P = 0.477) was observed between parasitemia and the variation in ALP activity.

**Association between the times elapsed since the appearance of the first symptoms before consultation and the levels of alteration of hepatic enzymes.** Table 7 presents the association between the levels of alterations in liver enzyme activity (normal, mild, moderate, and severe) and the time taken by febrile patients since the appearance of the first symptoms before consultation (< 3 days,} 3–7] days, and > 7 days). A significant association between the time taken before consultation and the alteration of serum ALT (Chi-square = 7.46; P = 0.024) and AST (Chi-square = 7.15; P = 0.027) activity was observed. However, no significant association (Chi-square = 4.82; P = 0.089) was found between the time taken before consultation and the alteration of ALP activity.

**Association between the place of treatment and the levels of alteration of hepatic enzymes.** Table 8 summarizes the statistics of associations between the levels of alteration in liver enzyme activity and the different places of care (Health center, self-medication, and herbalism). Overall, no significant association was found between the places of care and the level of alteration in AST (Chi-2 = 1.26; P = 0.532) or in ALP (Chi-2 = 4.13; P = 0.126) activity. In contrast, a significant association was observed between the places of treatment and the alteration of ALT activity (Chi-2 = 6.37; P = 0.041). The infection status, the parasitemia, the time taken

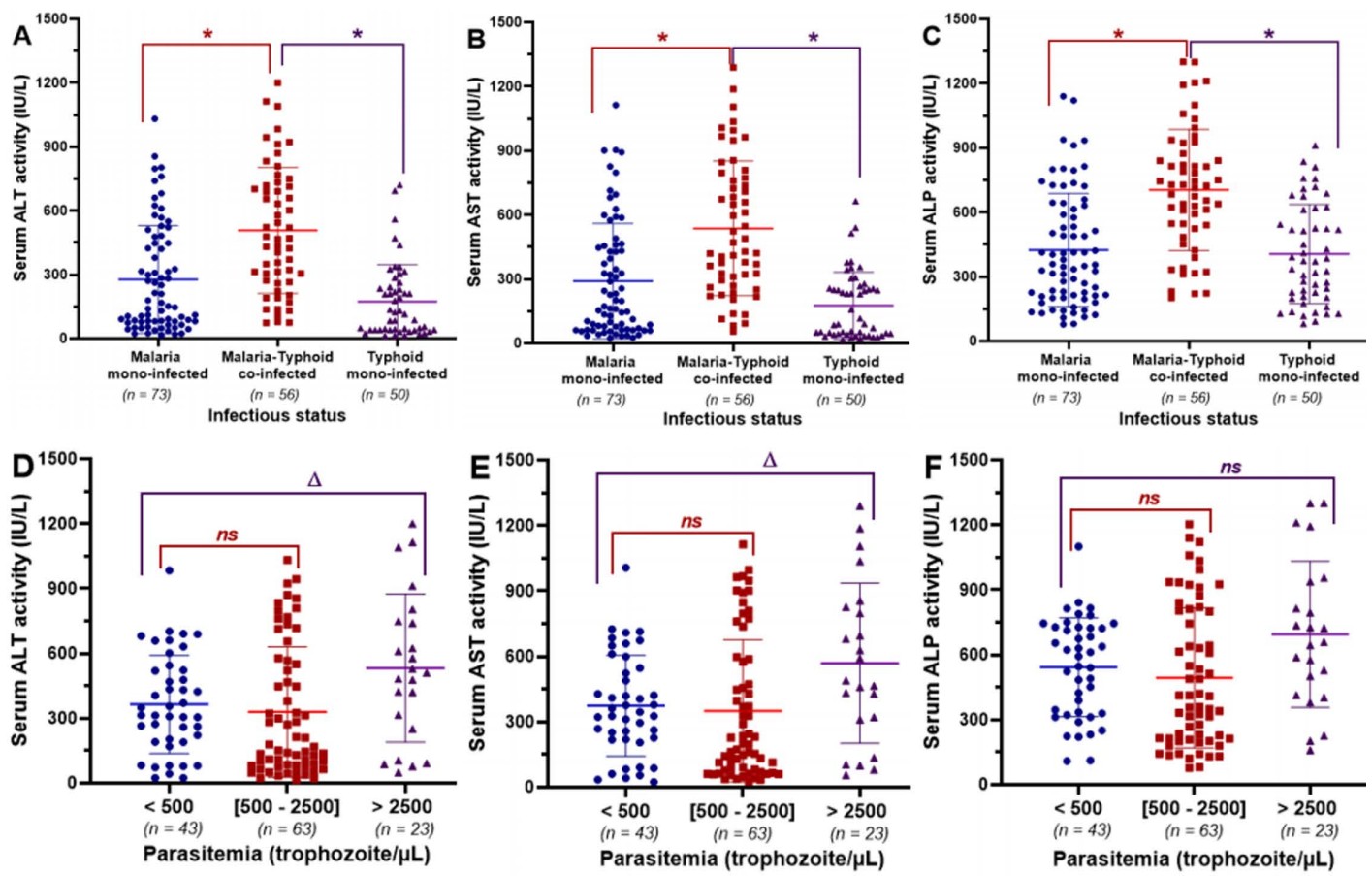

**Fig 2. Effect of the infection status and parasitemia on the variation of the biochemical markers of liver injury.** *(A) and (D), (B) and (E), (C) and (F): Effect of infection status and parasitemia on serum activities of ALT, AST, and ALP, respectively. Values are expressed as median and interquartile ranges and compared using non-parametric Mann Wihtney U test. * Median value significantly different when compared to co-infected patients (P<0.05). Δ Median value significantly different when compared to patients with a parasitemia lower than 500 parasites/µL of blood (P<0.05). ns Median value not significantly different when compared to patients with a parasitemia lower than 500 parasites/µL of blood (P>0.05). ALT: Alanine aminotransferase; AST: Aspartate aminotransferase; ALP: Alkaline phosphatase.*

before consultation, and place of care are considered as significant risk factors that can lead to the alterations in biochemical markers of liver injury.

### Risk factors related to the occurrence of liver injury in febrile patients suffering from malaria, typhoid fever, and malaria-typhoid co-infection

Table 9 presents the statistical associations between factors influencing the serum levels of liver enzymes (infection status, parasite density, time taken before consultation, and place of treatment) and the occurrence of different types of liver injuries (no injury, hepatocellular cytolysis, hepatic cholestasis, mixed lesion 'cytolysis/cholestasis') in malaria and/or typhoid fever infected patients.

The risk factors significantly associated with the occurrence of liver damage were respectively: infection status (Chi-squared = 18.03; P = 0.0001), time taken before consultation (Chi-squared = 13.29; P = 0.0013), and parasitemia (Chi-squared = 9.80; P = 0.0074). No significant association was found between the occurrence of liver damage and the place of treatment (Chi-squared = 2.78; P = 0.248).

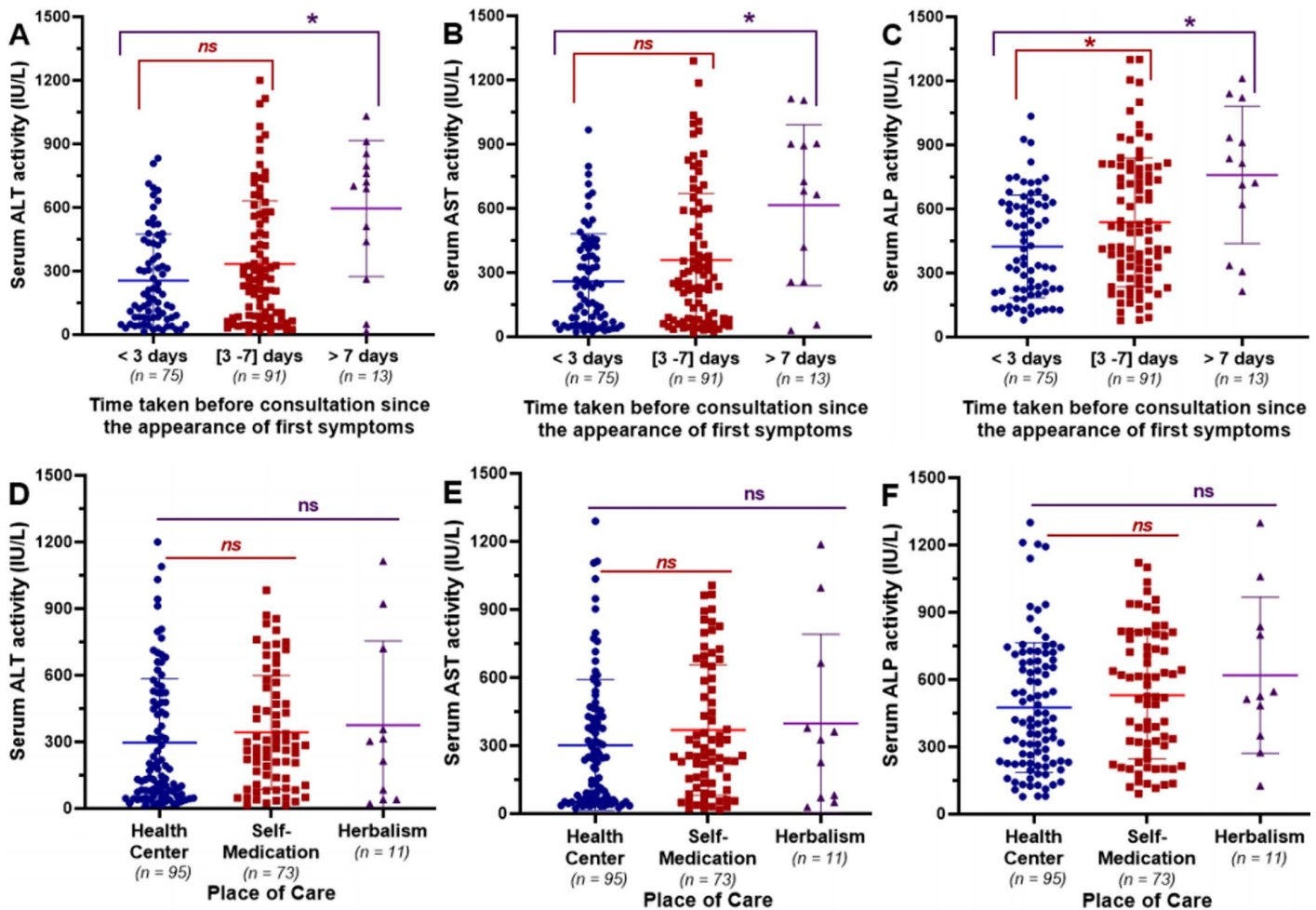

**Fig 3. Effect of the place of care and the time taken before consultation on the variation of the biochemical markers of liver injury.** *(A), (B), and (C): Effect of the time elapsed since the onset of the first symptoms before consultation on the serum activities of ALT, AST, and ALP, respectively. (D), (E), and (F): Effect of the initial place of care on the serum activities of ALT, AST, and ALP, respectively. Values are expressed as median and interquartile ranges and compared using non-parametric Mann Wihtney U test. * Median value significantly different when compared to patients with a consultation time of < 3 days (P<0.05). **ns** Median value not significantly different when compared to patients with a consultation time of < 3 days or those receiving care in a health center (P>0.05). ALT: Alanine aminotransferase; AST: Aspartate aminotransferase; ALP: Alkaline phosphatase.*

## Discussion

Malaria and typhoid fever continue to be endemic in tropical regions, particularly in sub-Saharan Africa, including Cameroon, where they represent a significant public health concern, causing hundreds of thousands of fatalities annually. Current diagnostic procedures and management approaches fail to evaluate liver damage in infected individuals. Even though the causative organisms, *Plasmodium* species for malaria and *Salmonella* serotypes for typhoid fever, have a critical hepatic phase in their pathogenic mechanisms, this aspect is often overlooked. Neglecting liver health can lead to adverse consequences that increase morbidity and mortality due to complications from liver injuries. Accordingly, continuous monitoring of biomarkers of liver injury in patients infected with malaria, typhoid, or both could help avert or lessen fatalities associated with liver damage. Thus, the present study aimed to identify the risk factors linked to liver damage in malaria, typhoid, and malaria-typhoid co-infected patients among febrile patients attending the CMA of Obili and the CHD

**Table 5. Association between the infection status and the levels of alteration of hepatic enzymes.**

| Variables | Categories | Infection status | | | Association | |
|---|---|---|---|---|---|---|
| | | Typhoid Mono-infection n (%) | Malaria Mono-infection n (%) | Malaria/Typhoid Co-infection n (%) | *Chi-²* | *P-value* |
| **ALT** | Severe | 20 (40.0) | 35 (47.9) | 46 (82.1) | 2.13 | 0.343 |
| | Moderate | 6 (12.0) | 11 (15.1) | 7 (12.5) | | |
| | Mild | 7 (14.0) | 18 (24.7) | 3 (5.4) | | |
| | Normal | 17 (34.0) | 9 (12.3) | 0 (0.0) | | |
| | **Total** | **50 (100)** | **73 (100)** | **56 (100)** | | |
| **AST** | Severe | 22 (44.0) | 34 (46.6) | 50 (89.2) | **8.44** | **0.014*** |
| | Moderate | 3 (6.0) | 12 (16.4) | 3 (5.4) | | |
| | Mild | 11 (22.0) | 20 (27.4) | 3 (5.4) | | |
| | Normal | 14 (28.0) | 7 (9.6) | 0 (0.0) | | |
| | **Total** | **50 (100)** | **73 (100)** | **56 (100)** | | |
| **ALP** | Severe | 13 (26.0) | 20 (27.4) | 38 (67.9) | **13.35** | **0.001*** |
| | Moderate | 20 (40.0) | 28 (38.4) | 14 (25.0) | | |
| | Mild | 15 (30.0) | 21 (28.8) | 4 (7.1) | | |
| | Normal | 2 (4.0) | 4 (5.5) | 0 (0.0) | | |
| | **Total** | **50 (100)** | **73 (100)** | **56 (100)** | | |

*The bold Chi-2 and P-value are indicators of a significant association*

of Mvog-Betsi in Yaoundé, while also emphasizing the patient's attitudes towards adherence to preventive strategies and the effective management of these conditions.

Out of the 350 feverish patients enrolled in this study, microscopic and serological diagnoses revealed an overall prevalence of 51.14% (197/350) for at least one infection. The infection rates of malaria and typhoid fever were 36.86% (129/350) and 30.29% (106/350), respectively, with a co-infection prevalence of 16.00% (56/350). Although the prevalence of malaria and typhoid observed in this study is lower than their respective prevalence reported by Kouam et al. (2023), who found a malaria prevalence of 75% (113/150) among febrile patients consulting at the Deido District Hospital in Douala, Cameroon, and by Nodem et al. (2023), who reported a typhoid fever prevalence of 64.3% (133/208) among febrile patients attending the regional hospital in the city of Ngaoundéré, the headquarters of the Adamawa region of Cameroon. These results, on one hand, confirm that malaria is still endemic in Cameroon, and the female Anopheles mosquitoes, the vectors responsible for its transmission, are proliferating everywhere in Cameroon, regardless of the city or region considered. Indeed, malaria accounts for nearly one-quarter of consultations and almost half of hospital admissions in Cameroon, with 2.9 million cases recorded in 2020, resulting in 4,000 fatalities [30]. On the other hand, the low prevalence of typhoid fever obtained in this study, compared to that of other Cameroonian cities such as Ngaoundéré, could be attributed to the more challenging access to good water quality in the city of Ngaoundéré, as compared to Yaoundé. It should be noted that the diagnosis of typhoid fever was based solely on the Widal test. Although still widely used in resource-limited countries including Cameroon, the Widal test has been criticized for having lower sensitivity and specificity compared to the isolation of *Salmonella typhi* bacteria through blood culture, which is considered the gold standard method for the diagnosis of typhoid fever [31]. This can, therefore, be regarded as a limitation of this study.

The transmission of malaria and typhoid fever is well known to be favored by deplorable socio-economic conditions [8,32]. Accordingly, this study analyzed the influence of socio-demographic characteristics on the rates of malaria and

**Table 6. Association between parasitemia and the levels of alteration of hepatic enzymes.**

| Variables | Categories | Parasitemia (Number of parasites/µL of blood) | | | Association | |
|---|---|---|---|---|---|---|
| | | < 500 n (%) | [500 - 2500] n (%) | > 2500 n (%) | Chi-² | P-value |
| **ALT** | Severe | 32 (74.4) | 31 (49.2) | 18 (78.3) | **9.36** | **0.009*** |
| | Moderate | 3 (7.0) | 14 (22.2) | 1 (4.3) | | |
| | Mild | 5 (11.6) | 12 (19.0) | 4 (17.3) | | |
| | Normal | 3 (7.0) | 6 (9.5) | 0 (0.0) | | |
| | **Total** | **43 (100)** | **63 (100)** | **23 (100)** | | |
| **AST** | Severe | 35 (81.4) | 31 (49.2) | 18 (78.3) | **13.75** | **0.001*** |
| | Moderate | 0 (0.0) | 13 (20.6) | 2 (8.7) | | |
| | Mild | 5 (11.6) | 15 (23.8) | 3 (13.0) | | |
| | Normal | 3 (7.0) | 4 (6.3) | 0 (0.0) | | |
| | **Total** | **43 (100)** | **63 (100)** | **23 (100)** | | |
| **ALP** | Severe | 21 (48.8) | 23 (36.5) | 14 (60.9) | 1.48 | 0.477 |
| | Moderate | 17 (39.5) | 19 (30.2) | 6 (26.1) | | |
| | Mild | 3 (7.0) | 19 (30.2) | 3 (13.0) | | |
| | Normal | 2 (4.7) | 2 (3.2) | 0 (0.0) | | |
| | **Total** | **43 (100)** | **63 (100)** | **23 (100)** | | |

*The bold Chi-2 and P-value are indicators of a significant association*

**Table 7. Association between the levels of alterations in liver enzyme activities and the time taken by febrile patients since the appearance of the first symptoms before consultation.**

| Variables | Categories | Time taken since the appearance of the first symptoms before consultation | | | Association | |
|---|---|---|---|---|---|---|
| | | < 3 days n (%) | [3–7] days n (%) | > 7 days n (%) | Chi-² | P-value |
| **ALT** | Severe | 35 (46.7) | 55 (60.4) | 11 (84.6) | **7.46** | **0.024*** |
| | Moderate | 15 (20.0) | 9 (9.9) | 0 (0.0) | | |
| | Mild | 13 (17.3) | 14 (15.4) | 1 (7.7) | | |
| | Normal | 12 (16.0) | 13 (14.3) | 1 (7.7) | | |
| | **Total** | **75 (100)** | **91 (100)** | **13 (100)** | | |
| **AST** | Severe | 37 (49.3) | 58 (63.7) | 11 (84.6) | **7.15** | **0.027*** |
| | Moderate | 12 (16.0) | 6 (6.6) | 0 (0.0) | | |
| | Mild | 14 (18.7) | 19 (20.9) | 1 (7.7) | | |
| | Normal | 12 (16.0) | 8 (8.8) | 1 (7.7) | | |
| | **Total** | **75 (100)** | **91 (100)** | **13 (100)** | | |
| **ALP** | Severe | 24 (32.0) | 37 (40.7) | 10 (76.9) | 4.82 | 0.089 |
| | Moderate | 25 (33.3) | 35 (38.5) | 2 (15.4) | | |
| | Mild | 23 (30.7) | 16 (17.6) | 1 (7.7) | | |
| | Normal | 3 (4.0) | 3 (3.3) | 0 (0.0) | | |
| | **Total** | **75 (100)** | **91 (100)** | **13 (100)** | | |

*The bold Chi-2 and P-value are indicators of a significant association*

**Table 8. Association between the levels of alteration in liver enzyme activities and the different places of care.**

| Variables | Categories | Places of care | | | Association | |
|---|---|---|---|---|---|---|
| | | Health Center n (%) | Self-Medication n (%) | Herbalism n (%) | Chi-² | P-value |
| **ALT** | Severe | 44 (46.3) | 50 (68.5) | 7 (63.6) | **6.37** | **0.041*** |
| | Moderate | 17 (17.9) | 7 (9.6) | 0 (0.0) | | |
| | Mild | 17 (17.9) | 10 (13.7) | 1(9.1) | | |
| | Normal | 17 (17.9) | 6 (8.2) | 3 (27.3) | | |
| | **Total** | **95 (100)** | **73 (100)** | **11 (100)** | | |
| **AST** | Severe | 50 (52.6) | 49 (47.9) | 7 (63.6) | 1.26 | 0.532 |
| | Moderate | 9 (9.5) | 9 (12.3) | 0 (0.0) | | |
| | Mild | 22 (23.2) | 9 (12.3) | 1(9.1) | | |
| | Normal | 14 (14.7) | 6 (8.2) | 3 (27.3) | | |
| | **Total** | **95 (100)** | **73 (100)** | **11 (100)** | | |
| **ALP** | Severe | 32 (33.7) | 35 (47.9) | 4 (36.4) | 4.13 | 0.126 |
| | Moderate | 36 (37.9) | 20 (27.4) | 6 (54.5) | | |
| | Mild | 22 (23.2) | 17 (23.3) | 1 (9.1) | | |
| | Normal | 5 (5.3) | 1 (1.4) | 0 (0.0) | | |
| | **Total** | **95 (100)** | **73 (100)** | **11 (100)** | | |

*The bold Chi-2 and P-value are indicators of a significant association*

typhoid infection. Our results showed that men were more affected than women by both malaria and typhoid fever. This observation could be justified by the fact that men are more exposed to mosquito bites compared to women, likely because they tend to spend more time outside due to their daily activities. Additionally, given that one of the major routes of typhoid fever transmission is through the ingestion of contaminated water and/or food [8], men are at a higher risk of contracting Salmonella infection because they are more likely to consume roadside food where adherence to basic hygiene rules is not always respected during the cooking process [33]. It was also observed that patients aged above 60 years were less affected by malaria or typhoid fever when compared to patients in other age groups. This could be explained by the fact that older people may have a better knowledge about the diseases and are more likely to strictly follow the preventive measures compared to younger patients. In addition, patients living in houses where pit latrines are used as a model of sanitation were significantly at risk (RR = 1.92; P < 0.0001) of being exposed to typhoid fever compared to those using water system toilets. Indeed, pit latrines are naturally unsanitary and are more susceptible to facilitating the fecal -oral transmission of Salmonella species. Furthermore, the use of tap water as a source of running or drinking water was significantly (RR = 2.38; P < 0.0001) associated with Salmonella infection. Considering the waterborne nature of typhoid fever, a treatment step such as filtration or boiling of tap water, and even spring, well, and borehole water prior to use could be beneficial in avoiding typhoid fever in endemic areas.

Despite being caused by different pathogens and transmitted in various ways, malaria and typhoid fever exhibit similar symptoms [34]. Accordingly, the associations between the clinical symptoms experienced by febrile patients and the infection rate of each disease were assessed. Febrile patients presenting clinical symptoms such as fever, headaches, chills, and muscle aches were significantly at risk (RR > 2.66; P < 0.0001) of being infected by malaria parasites, while abdominal pain, asthenia, and vomiting were significantly (RR > 2.14; P < 0.0001) associated with typhoid fever. These observations corroborate previous findings that report gastrointestinal complaints like abdominal pain and vomiting are predominantly seen in typhoid fever, while persistent fever, headaches, muscle pain, and chills are the most frequent symptoms found

in malaria-infected patients [33–35]. However, it is important to mention that only 51.14% (179/350) of the febrile patients enrolled in the study were diagnosed with either malaria or typhoid fever. This observation highlights the need for proper diagnosis before any treatment, despite the presence of clinical symptoms, to avoid self-medication or misuse of drugs, which can exacerbate the patient's condition

To better understand the reason behind the persistent high prevalence of typhoid fever and malaria among febrile patients, their attitudes toward adherence to preventive measures were assessed. This included their compliance with vector control strategies, such as maintaining a clean environment near their homes, using long-lasting treated mosquito nets, applying mosquito spray insecticides, and utilizing intermittent prophylaxis for malaria control; as well as adhering to basic hygiene rules, including hand washing before meals, washing fruits and vegetables before consumption, and proper garbage disposal for typhoid control. Results from this study revealed that up to 28% (99/350), 16.3% (50/350), 32.3% (113/350), and 37.7% (152/350) of the febrile patients stayed out late at night, had standing-water near their homes, had bushes near their homes, and were irregular or non-users of mosquito nets, respectively. Coincidentally, they were significantly at risk (RR>2; P<0.0001) of being infected by *Plasmodium* species. Indeed, risky behaviors such as staying outside late at night, retaining

**Table 9. Association between factors influencing the serum levels of liver enzymes activities and the occurrence of different types of liver injuries.**

| Variables | Categories | Type of hepatic injuries | | | | Association | |
|---|---|---|---|---|---|---|---|
| | | Mixed n (%) | Cholesta-sis n (%) | Hepato-cellular n (%) | No injury n (%) | Chi-² | P-value |
| Infection status | Co-infection Malaria/ Typhoid -56) n (%) | 29 (51.8) | 23 (41.1) | 2 (3.6) | 2 (3.6) | **18.03** | **0.0001*** |
| | Mono-infection Malaria (N=73) n (%) | 28 (38.4) | 19 (26.0) | 10 (13.7) | 16 (21.9) | | |
| | Mono-infection Typhoid (N=50) n (%) | 5 (10.0) | 28 (56.0) | 2 (4.0) | 15 (30.0) | | |
| Parasitemia (Trophozoite/µL) | > 2500 (N=23) n (%) | 15 (65.2) | 5 (21.7) | 1 (4.3) | 2 (8.7) | **9.80** | **0.0074*** |
| | [500–2500] (N=63) n (%) | 28 (44.4) | 14 (22.2) | 9 (14.3) | 12 (19.0) | | |
| | < 500 (N=43) n (%) | 14 (32.6) | 23 (53.5) | 2 (4.7) | 4 (9.3) | | |
| Time taken before consultation | > 7 days (N=13) n (%) | 11 (84.6) | 1 (7.7) | 0 (0.0) | 1 (7.7) | **13.29** | **0.0013*** |
| | [3–7] days (N=91) n (%) | 35 (38.5) | 37 (40.7) | 4 (4.4) | 15 (16.5) | | |
| | < 3 days (N=75) n (%) | 16 (21.3) | 32 (42.7) | 10 (13.3) | 17 (22.7) | | |
| Place of care | Herbalism (N=11) n (%) | 3 (27.3) | 7 (63.6) | 0 (0.0) | 1 (9.1) | 2.78 | 0.248 |
| | Self-Medication (N=73) n (%) | 30 (41.1) | 25 (34.2) | 6 (8.2) | 12 (16.4) | | |
| | Health Center (N=95) n (%) | 29 (30.5) | 38 (40.0) | 8 (8.4) | 20 (21.1) | | |

The bold Chi-2 and P-value are indicators of a significant association

standing water or bushes near the home, and the inconsistent use of mosquito nets not only enhance the proliferation of vectors but also facilitate mosquito bites. Similar observations have been reported in previous studies conducted in Cameroon and Nigeria, two countries where malaria remains a major public health concern [6,36]. These findings indicate that a considerable proportion of the population living in malaria - endemic zones does not place sufficient value on adhering to the preventive measures necessary to counteract the transmission and propagation of malaria. Accordingly, more community health campaigns should be conducted in these endemic areas to continually enhance awareness about the effective implementation of preventive measures to combat malaria. Likewise, we observed that the presence of garbage dumps near residences were significantly (RR=2.34; P<0.0001) associated with typhoid fever. Indeed, garbage dumps are breeding grounds for the proliferation of bacteria such as *Salmonella* species, and their proximity to people's residences can facilitate the contamination of water and food, which subsequently transmits the disease through ingestion. Accordingly, maintaining a clean environment around residences is necessary to reduce the risk of infection by *Salmonella* species.

The pathogens responsible for malaria or typhoid fever have a compulsory hepatic phase throughout their infection life cycle in humans [11–14], which can affect the liver, a key organ that performs numerous vital functions for the optimal functioning of the body [6,15,16]. In fact, *Plasmodium* and *Salmonella* species can invade liver cells, leading to organ congestion and inflammation that ultimately results in cellular necrosis and cholestasis, characterized by an abnormal alteration of serum levels of liver enzyme activities, including transaminases: alanine aminotransferase (ALT) and aspartate aminotransferase (AST), along with alkaline phosphatase (ALP) [7,17,18]. The elevation of serum levels of liver enzyme activity has been reported in febrile patients suffering from malaria or typhoid fever [15,37]. Furthermore, dual infections by *Plasmodium* and *Salmonella* species are often reported in endemic areas [29,38]. As a consequence, in cases of co-infection, both *Salmonella* and *Plasmodium* species can damage liver cells, either simultaneously or successively, depending on the chronological order of infection, thus exacerbating liver impairment [38]. In this study, we found that independently of the infection status (malaria mono-infected, typhoid mono-infected, or malaria-typhoid co-infected) of the patients, the level of alteration of liver enzyme activities varies from mild to severe increases. However, the serum activities of these enzymes were significantly (P<0.05) elevated in co-infected patients compared to those with malaria or typhoid mono-infections. This observation suggests that co-infection is more likely to cause severe liver injuries than mono-infection, although the serum levels of these liver enzyme activity were at least two times greater than the upper limit of the normal range of their respective reference values. Regarding the malaria-infected patients only, we observed that patients with parasite density above 2500 trophozoites/μL displayed a significant (P<0.05) increase in serum levels of liver enzyme activities compared to those with parasite density <2500 trophozoites/μL. These results suggest that parasitemia is a risk factor that negatively affects the liver. In fact, high parasite density associated with severe forms of malaria and hepatic failure, marked by increased serum levels of liver enzyme activity, have been reported [22,39]. Based on the values of ALT and ALP activities, the types of liver injuries related to their levels of alterations were defined according to the RUCAM classification as follows: Hepatocellular, Cholestasis, and a mixed pattern of cholestasis and hepatocellular damages. The Chi- squared analysis revealed that the infection status (Chi-$^2$=18.03; P<0.0001) and parasite density (Chi-$^2$=9.80; P=0.0074) were significant risk factors for liver damages. Moreover, 51.8% (29/56) of co-infected patients were more likely to develop mixed injuries compared to malaria mono-infected (38.4%; 28/73) or typhoid mono-infected (10%; 5/10) patients. Likewise, 65.2% (15/23) of malaria-infected patients with a parasitemia greater than 2500 trophozoites/μL displayed a mixed injury, compared to 44.4% (28/63) and 32.6% (14/43) for patients with parasite densities between [500–2500] or less than 500 trophozoites/μL respectively. These findings suggest that high parasite density is a significant risk factor for developing severe liver injury and potential liver failure, which may be fatal to patients.

Consulting in an authorized health center immediately after the onset of the first symptoms is crucial for effective treatment of malaria and typhoid fever. Additionally, it has been shown that delays in seeking medical care after the initial symptoms of malaria or typhoid fever increase the risk of developing severe liver injury, characterized by high serum levels of liver enzyme activity [6,38–40]. In this study, we observed that infected patients who spent more than seven days before

seeking medical care exhibited a significant (P<0.05) increase in serum levels of liver enzyme activity compared to those who consulted before three days or between three and seven days. Regarding the initial place of treatment: health center, self-medication, or herbalism, no significant difference (P>0.05) was observed in the serum levels of liver enzyme activity, although the serum levels of these liver enzyme activities were at least two times greater than the upper limit of the normal range of their respective reference values for each group considered. These findings suggest that the serum levels of liver enzyme activity and the potential types of liver damage observed in infected patients are related to the time taken before consultation after the appearance of the first symptoms, whereas this is not the case with the place of treatment. These observations were confirmed by the Chi- squared analysis, which showed a significant association (Chi-²=13.29; P=0.0013) between the time elapsed before consultation and the occurrence of liver damage, and a non-significant association (Chi-²=2.78; P=0.248) between the place of treatment and the occurrence of liver damage. Furthermore, patients with a time before consultation greater than seven days (84.6%; 11/13) were more likely to develop mixed injury compared to those consulting between three to seven days (38.5%; 35/91) or within three days (21.3%; 16/75) after the onset of the first symptoms. Based on these findings, it is recommended that febrile patients refer immediately to an appropriate health center for medical attention as soon as they manifest clinical symptoms of malaria or typhoid fever. This could be helpful in preventing the occurrence of severe liver injuries often associated with the high mortality rate of malaria and typhoid fever.

## Conclusion

As we reach the conclusion of this study, which aimed to determine the type of liver damage and associated risk factors in malaria, typhoid fever, or malaria-typhoid co-infected patients among febrile individuals consulting at the Obili District Medical Center (CMA) and the Dominican Hospital Center (CHD) Saint Martin de Porres in Mvog-Betsi, Yaoundé, Cameroon, it can be concluded that adherence to preventive measures against malaria and typhoid fever was partially observed among the febrile patients. Furthermore, the infection rates were linked to risk factors such as unsanitary living conditions and the non- use of mosquito nets. Additionally, high parasite density, the malaria-typhoid co- infection status, and the delay in consulting after the appearance of the first symptoms of the diseases represented major risk factors associated with alterations in serum levels of liver enzyme activity and the occurrence of liver damages. Based on these findings, we therefore recommend that the population fully implement preventive measures against malaria and typhoid fever daily and seek medical attention immediately upon the appearance of the first symptoms of the diseases. Moreover, healthcare personnel should prescribe liver function tests to malaria, typhoid fever, or malaria-typhoid co-infected patients, especially for those with high parasite density or those attending the hospital more than seven days after the onset of the disease. This could be useful in detecting early signs of liver damage and prompting medical intervention to prevent further complications that may be fatal for the patients.

## Supporting information

**S1 File. Consent form and structured questionnaire.**
(DOCX)

**S2 File. Supplementary tables.**
(DOCX)

**S3 File. General database.**
(XLSX)

## Acknowledgments

The authors express their gratitude to all participants involved in this study and acknowledge the laboratory staff at the Obili District Medical Center for their technical assistance.

## Author contributions

**Conceptualization:** Arnaud Fondjo Kouam, Paul Fewou Moundipa, Frédéric Nico Njayou.

**Data curation:** Madeleine Yvanna Nyangono Essam, Arnaud Fondjo Kouam, Armelle Gaelle Kwesseu Fepa, Elisabeth Menkem Zeuko'o, Felicité Syntia Douanla Somene, Nembu Erastus Nembo, Frédéric Nico Njayou.

**Formal analysis:** Madeleine Yvanna Nyangono Essam, Arnaud Fondjo Kouam, Armelle Gaelle Kwesseu Fepa, Armel Jackson Seukep, Elisabeth Menkem Zeuko'o, Felicité Syntia Douanla Somene, Nembu Erastus Nembo, Frédéric Nico Njayou.

**Funding acquisition:** Arnaud Fondjo Kouam.

**Investigation:** Madeleine Yvanna Nyangono Essam, Arnaud Fondjo Kouam, Armelle Gaelle Kwesseu Fepa.

**Methodology:** Madeleine Yvanna Nyangono Essam, Arnaud Fondjo Kouam, Armelle Gaelle Kwesseu Fepa.

**Resources:** Paul Fewou Moundipa, Frédéric Nico Njayou.

**Software:** Madeleine Yvanna Nyangono Essam, Arnaud Fondjo Kouam, Elisabeth Menkem Zeuko'o, Nembu Erastus Nembo.

**Supervision:** Arnaud Fondjo Kouam, Armel Jackson Seukep, Elisabeth Menkem Zeuko'o, Paul Fewou Moundipa, Frédéric Nico Njayou.

**Validation:** Armel Jackson Seukep, Elisabeth Menkem Zeuko'o, Felicité Syntia Douanla Somene, Nembu Erastus Nembo, Paul Fewou Moundipa, Frédéric Nico Njayou.

**Visualization:** Armel Jackson Seukep, Elisabeth Menkem Zeuko'o, Felicité Syntia Douanla Somene, Nembu Erastus Nembo, Frédéric Nico Njayou.

**Writing – original draft:** Madeleine Yvanna Nyangono Essam, Arnaud Fondjo Kouam, Armelle Gaelle Kwesseu Fepa.

**Writing – review & editing:** Madeleine Yvanna Nyangono Essam, Arnaud Fondjo Kouam, Armelle Gaelle Kwesseu Fepa, Armel Jackson Seukep, Elisabeth Menkem Zeuko'o, Felicité Syntia Douanla Somene, Nembu Erastus Nembo, Paul Fewou Moundipa, Frédéric Nico Njayou.

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
