## [Decision Letter · Decision Letter 0]

19 Mar 2025

Dear Dr. KOUAM,

We look forward to receiving your revised manuscript.

Kind regards,

Hope Onohuean, PhD

Academic Editor

PLOS ONE

Journal Requirements:

2. Please remove all personal information, ensure that the data shared are in accordance with participant consent, and re-upload a fully anonymized data set.

Reviewers' comments:

Reviewer's Responses to Questions

**Comments to the Author**

1. Is the manuscript technically sound, and do the data support the conclusions?

Reviewer #1: Partly

Reviewer #2: Yes

2. Has the statistical analysis been performed appropriately and rigorously?

Reviewer #1: Yes

Reviewer #2: I Don't Know

3. Have the authors made all data underlying the findings in their manuscript fully available?

Reviewer #1: Yes

Reviewer #2: Yes

4. Is the manuscript presented in an intelligible fashion and written in standard English?

Reviewer #1: Yes

Reviewer #2: Yes

Reviewer #1: The authors set out " to determine the type of liver damage and the associated risk factors in malaria, typhoid fever, or malaria-typhoid co-infected patients among febrile patients" in a cohort of350 patients seen at two hospitals in Yaoundé, Cameroon. Though not clearly articulated the working hypothesis seems to be that since the causative agents of malaria and typhoid fever pass through a liver stage they may case liver damage which is often neglected in the management of these diseases ,and may result in poor clinical outcomes or even fatalities.

To test these hypothesis the authors collected blood from febrile patients, carried out serologic diagnosis of Salmonellosis ( typhoid fever) and microscopic diagnosis of malaria using Giemsa stained thin and thick smears. Further more they measured the serum levels the enzymes ALT, ALP, AST whose elevation above normal levels generally signal liver damage. It is regrettable that the authors did not recruit a control group of study participants matched by age and sex for comparison. The authors did not mention how fever was determined, and whether this was done at the point of blood collection. This information should be given in the Methods Section.

Using a questionnaire the authors recorded a number of socio-demographic parameters as well as environmental factors that could influence the transmission of the two diseases they studied. The authors then generated copious data which they analysed using standard statistical packages. Their analyses showed correlations between the elevation of liver enzymes and the infections studied. However, the lack of proper controls makes the conclusions less convincing.

The paper is generally well written and well illustrated. However the discussion is too long and repetitive. The discussion should be shortened focus on interpreting the results rather than summarizing or restating them. The conclusion sounds like a summary. It should bring out the key finding/recommendation of the study in a sentence or two.

I found a few stylistic errors which need to be corrected in the revised paper as follows:

line 72: write '..million case..."instead of ... 'instances.

line 11-92 Write: 'Similarly...'instead of 'Conversely...'

line 161: Write washing of fruits... instead of 'hygiene...'

line 94: Write: '... liver damage instead of 'liver damages'.

line 208: Give the meaning of ULN in full. Does it mean Upper Normal limit?

lin1 247: SI and S2 should be in the main text not in the supplementary results' section

Overall Recommendation: accept after major revisions.

Reviewer #2: REVIEWERS RECOMMENDATION

Serological evidence and factors associated to liver damage in malaria-typhoid infected patients consulting in two health facilities, Yaoundé-Cameroon.

This study focused on an often-neglected area of the impact of malaria and typhoid on the liver function of infected patients in two clinics in Yaoundé-Cameroon. The authors identify risk factors for infections and liver injury by using questionnaires, microscopy (malaria), Widal tests (typhoid), and liver enzyme assays. Key findings include higher infection rates in males, environmental risk factors (e.g., standing water), and co-infection/delayed consultation as significant predictors of liver damage. This is good research that addresses a critical gap in understanding liver injury in Africa. However, there are concerns,

• The abstract (lines 47-48) says, “Men (58/118 and 44/118) were more affected than 48 women (71/232 and 62/232) for both malaria and typhoid”. I believe using percentage here will communicate your result better than the proportion.

• The Widal test used for determining typhoid has been criticized for having low sensitivity and specificity. Can another confirmatory test be used to establish typhoid in the patients, or do you acknowledge this limitation in your discussion?

• The study needs a control group of non-febrile individuals to compare the baseline liver enzyme levels in the general population.

• There is a need to consider other cofounders, like previous or active viral infections such as hepatitis, that can further influence liver enzyme levels. Also, there is a need to analyze the potential confounding effect of drug-induced liver injury versus pathogen-induced liver damage. Statistically, you can also perform multivariate analysis to adjust for confounders.

• In the method section, there is no clear description of how consent from underage patients was obtained.

• While the Roussel Uclaf Causality Assessment Method (RUCAM) is an acceptable tool, it is typically used for drug-induced liver injury. Its applicability to infectious diseases should be justified.

• Some thresholds used in the method section were not clearly defined. For example, lines 205-206, “In our laboratory, these reference values in humans are: from 10 to 42 UI/L, from 8 to 39 UI/L, and from 40 to 129 UI/L respectively for ALT, AST, and ALP.” How does your reference value compare to those used in other countries, and how did you arrive at your reference values? Also, the threshold for classification of parasite density (low: <500, medium: 500–2500, high: >2500 parasites/μL) should be backed with references

• The figures are not up to standard. Please improve the quality of your graphs.

There are a lot of grammatical errors on the document, and I will recommend a thorough revision of the manuscript. Some of the ones spotted are listed below:

• Line 38 - "risk factors associated to" should be "risk factor associated with".

• Line 40 - "During 8 months" should be "Over 8 months"

• Line 43 - "Liver damage was assessed employing" should be “Liver damage was assessed using"

• Line 54 - "Liver enzyme activities, reflecting liver damages" should be "Liver enzyme activity, reflecting liver damage"

• Line 56 - "should be considered during the patient's treatment." should be "should be considered during patient treatment."

In the discussion, there are several sentences that needs to be splitted into clearer sentences. For example; “However, current diagnostic procedures and management approaches do not consider evaluating liver damage in infected individuals, even though their causative organisms, Plasmodium species for malaria and Salmonella serotypes for typhoid fever, have a necessary hepatic phase in their respective pathogenic mechanisms, which is not without

adverse consequences on the liver, and this can increase the morbidity and mortality as a result of complications arising from liver injuries.”

**Do you want your identity to be public for this peer review?** For information about this choice, including consent withdrawal, please see our Privacy Policy

Reviewer #1: No

Reviewer #2: No

---

## [Author Response · Author response to Decision Letter 0]

28 Mar 2025

RESPONSE TO THE REVIEWER’S COMMENTS

Note: All changes or suggestions are written in red in the revised manuscript

Dear Editor-in-Chief of the journal: PLOS ONE

Thank you for your letter dated 20th March 2025 and the opportunity given to us to revise and resubmit the manuscript entitled “Serological evidence and factors associated to liver damage in malaria-typhoid infected patients consulting in two health facilities, Yaoundé-Cameroon. PONE-D-25-05617”. We would also like to take this opportunity to thank the editorial team and reviewers for their helpful comments, which greatly contributed to improving the current version of this manuscript. The manuscript has been updated in accordance with the reviewers' recommendations. Most of their inquiries have been answered, and some clarifications have been provided. Throughout this revised manuscript, modifications are written in red.

Editor and Reviewer comments

Editor’s comment

Comment #1: Please ensure that your manuscript meets PLOS ONE's style requirements, including those for file naming. The PLOS ONE style templates can be found at https://journals.plos.org/plosone/s/file?id=wjVg/PLOSOne_formatting_sample_main_body.pdf and https://journals.plos.org/plosone/s/file?id=ba62/PLOSOne_formatting_sample_title_authors_affiliations.pdf

Author Response: Dear Editor, thank you for your remark. The name of all files associated to this submission has been revised according to PLOS ONE’s style requirements.

Comment #2: Please remove all personal information, ensure that the data shared are in accordance with participant consent, and re-upload a fully anonymized data set.

Author Response: Dear Editor, thank you for your suggestions. We have removed all personal information. Also, the spreadsheet columns do not contain any personal information and the patient’s name has been replaced with code.

Reviewer #1:

The authors set out " to determine the type of liver damage and the associated risk factors in malaria, typhoid fever, or malaria-typhoid co-infected patients among febrile patients" in a cohort of 350 patients seen at two hospitals in Yaoundé, Cameroon. Though not clearly articulated the working hypothesis seems to be that since the causative agents of malaria and typhoid fever pass through a liver stage they may cause liver damage which is often neglected in the management of these diseases ,and may result in poor clinical outcomes or even fatalities.

To test these hypothesis the authors collected blood from febrile patients, carried out serologic diagnosis of Salmonellosis ( typhoid fever) and microscopic diagnosis of malaria using Giemsa stained thin and thick smears. Furthermore, they measured the serum levels the enzymes ALT, ALP, AST whose elevation above normal levels generally signal liver damage. It is regrettable that the authors did not recruit a control group of study participants matched by age and sex for comparison. The authors did not mention how fever was determined, and whether this was done at the point of blood collection. This information should be given in the Methods Section.

Using a questionnaire the authors recorded a number of socio-demographic parameters as well as environmental factors that could influence the transmission of the two diseases they studied. The authors then generated copious data which they analyzed using standard statistical packages. Their analyses showed correlations between the elevation of liver enzymes and the infections studied. However, the lack of proper control makes the conclusions less convincing.

Comment #1: The paper is generally well written and well-illustrated. However, the discussion is too long and repetitive. The discussion should be shortened to focus on interpreting the results rather than summarizing or restating them. The conclusion sounds like a summary. It should bring out the key finding/recommendation of the study in a sentence or two.

Author Response: Dear Reviewer, thank you for your suggestions. Regarding the discussion section, we agree that it seems to be long. Our intention was to emphasis on each factor that could increase the malaria infection rate or represent a risk of liver damage. However, efforts have been made to shorten the length and concentrate more on interpretating the results. Concerning the conclusion, key findings have already been included, and recommendations have been added. Modification is as follows: Based on these findings, we therefore recommend that the population fully implement preventive measures against malaria and typhoid fever daily and seek medical attention immediately upon the appearance of the first symptoms of the diseases. Moreover, healthcare personnel should prescribe liver function tests to malaria, typhoid fever, or malaria-typhoid co-infected patients, especially for those with high parasite density or those attending the hospital more than seven days after the onset of the disease. This could be useful in detecting early signs of liver damage and prompting medical intervention to prevent further complications that may be fatal for the patients.

I found a few stylistic errors which need to be corrected in the revised paper as follows:

Comment #2: line 72: write '…million cases..."instead of ... 'instances.

Author Response: Dear Reviewer, thank you for your suggestion. It has been corrected in the revised manuscript.

Comment #3: line 11-92 Write: 'Similarly...'instead of 'Conversely...'

Author Response: Dear Reviewer, thank you for your remark. It has been corrected in the revised manuscript.

Comment #4: line 161: Write washing of fruits... instead of 'hygiene...'

Author Response: Dear Reviewer, thank you for your suggestion. It has been corrected in the revised manuscript.

Comment #5: line 94: Write: '... liver damage instead of 'liver damages'.

Author Response: Dear Reviewer, thank you for your suggestion. It has been corrected in the revised manuscript.

Comment #6: line 208: Give the meaning of ULN in full. Does it mean Upper Normal limit?

Author Response: Dear Reviewer, thank you for your remark. The full meaning of ULN has been provided and it is highlighted in red in the revised manuscript. ULN stands for Upper Limit of the Normal range of values.

Comment #7: line 247: SI and S2 should be in the main text not in the supplementary results' section.

Author Response: Dear Reviewer, thank you for your remark. Supplementary Tables S1 and S2 present the frequency distribution of the socio-demographic characteristics and clinical symptoms of febrile patients enrolled in the study, respectively. We agree that these two tables can be included in the main text. However, to limit the number of tables appearing in the main text, we believe it is convenient to keep these tables in the supplementary file. Nevertheless, similar information can be seen in Tables 1 and 2 included in the main text.

Reviewer #2: REVIEWERS RECOMMENDATION

Serological evidence and factors associated to liver damage in malaria-typhoid infected patients consulting in two health facilities, Yaoundé-Cameroon. This study focused on an often-neglected area of the impact of malaria and typhoid on the liver function of infected patients in two clinics in Yaoundé-Cameroon. The authors identify risk factors for infections and liver injury by using questionnaires, microscopy (malaria), Widal tests (typhoid), and liver enzyme assays. Key findings include higher infection rates in males, environmental risk factors (e.g., standing water), and co-infection/delayed consultation as significant predictors of liver damage. This is good research that addresses a critical gap in understanding liver injury in Africa. However, there are concerns,

Comment #1: The abstract (lines 47-48) says, “Men (58/118 and 44/118) were more affected than 48 women (71/232 and 62/232) for both malaria and typhoid”. I believe using percentage here will communicate your result better than the proportion.

Author Response: Dear Reviewer, thank you for your suggestion. It has been corrected in the revised manuscript.

Comment #2: The Widal test used for determining typhoid has been criticized for having low sensitivity and specificity. Can another confirmatory test be used to establish typhoid in patients, or do you acknowledge this limitation in your discussion?

Author Response: Dear Reviewer, thank you for your comment. We agree that the Widal test has been criticized for having low sensitivity and specificity. However, it is still widely used for the serological diagnosis of typhoid fever, especially in resource-limited countries like Cameroon. In this study, we were unable to perform a confirmatory test, such as isolating Salmonella typhi bacteria through blood culture, considered the gold standard method for diagnosing typhoid fever, due to financial constraints. However, as you suggested, this limitation has been acknowledged in the discussion section.

Comment #3: The study needs a control group of non-febrile individuals to compare the baseline liver enzyme levels in the general population.

Author Response: Dear Reviewer, thank you for your comment. We agree that it would have been better to include a control group of non-febrile individuals to compare the baseline liver enzyme levels in the general population. However, the comparison was made based on the lower and upper limits of the normal range of values for an individual in good health for each parameter evaluated, as indicated in the instructions manual of the assay kit used. That is why we did not include the control group of non-febrile individuals.

Comment #4: There is a need to consider other co-founders, like previous or active viral infections such as hepatitis, that can further influence liver enzyme levels. Also, there is a need to analyze the potential confounding effect of drug-induced liver injury versus pathogen-induced liver damage. Statistically, you can also perform multivariate analysis to adjust for confounders.

Author Response: Dear Reviewer, thank you for your comment. We agree that it is important to consider co-founders while evaluating liver enzyme levels, as they can significantly influence the results of the test. To limit bias in our study, patients with a history of or active viral hepatitis infections or patients on known hepatotoxic drugs were excluded from the study, as indicated in the paragraph "Study Design and Target Population".

Comment #5: In the method section, there is no clear description of how consent from underage patients was obtained.

Author Response: Dear Reviewer, thank you for your comment. The procedure to obtain consent from parents or guardians of underage patients was the same as that for adult patients. This is indicated in the paragraph titled "Sampling Technique and Estimation of Sample Size".

Comment #6: While the Roussel Uclaf Causality Assessment Method (RUCAM) is an acceptable tool, it is typically used for drug-induced liver injury. Its applicability to infectious diseases should be justified.

Author Response: Dear Reviewer, thank you for your comment. We agree that the RUCAM method is primarily utilized for drug-induced liver injury. Since the parameters used in the RUCAM method (ALT and ALP) were also assessed in this study, our intention was to apply the same method to gain insight into the type of liver damage that may occur in patients with malaria, typhoid fever, or malaria-typhoid co-infection.

Comment #7: Some thresholds used in the method section were not clearly defined. For example, lines 205-206, “In our laboratory, these reference values in humans are: from 10 to 42 UI/L, from 8 to 39 UI/L, and from 40 to 129 UI/L respectively for ALT, AST, and ALP.” How does your reference value compare to those used in other countries, and how did you arrive at your reference values? Also, the threshold for classification of parasite density (low: <500, medium: 500–2500, high: >2500 parasites/μL) should be backed with references.

Author Response: Dear Reviewer, thank you for your comment. The reference values used in this study were adapted from those found in the manual instructions of the assay kits, all purchased from BIOLABO, Les Hautes Rives, Maizy, France, and the reference values indicated are comparable to those applied in other countries. Regarding the threshold for classifying parasite density, an appropriate reference has been provided.

•The figures are not up to standard. Please improve the quality of your graphs.

Author Response: Dear Reviewer, thank you for your remark. The figures have been provided in high resolution. The low quality observed in the PDF used for review is due to the online submission system, which often decreases the quality of some images when generating the PDF file intended for review purposes.

Comment #8: There are a lot of grammatical errors on the document, and I will recommend a thorough revision of the manuscript. Some of the ones spotted are listed below:

• Line 38 - "risk factors associated to" should be "risk factor associated with".

• Line 40 - "During 8 months" should be "Over 8 months"

• Line 43 - "Liver damage was assessed employing" should be “Liver damage was assessed using"

• Line 54 - "Liver enzyme activities, reflecting liver damages" should be "Liver enzyme activity, reflecting liver damage"

• Line 56 - "should be considered during the patient's treatment." should be "should be considered during patient treatment."

Author Response: Dear Reviewer, thank you for your remarks. Grammatical errors have been checked and corrected in the revised manuscript.

Comment #9: In the discussion, there are several sentences that need to be splitted into clearer sentences. For example; “However, current diagnostic procedures and management approaches do not consider evaluating liver damage in infected individuals, even though their causative organisms, Plasmodium species for malaria and Salmonella serotypes for typhoid fever, have a necessary hepatic phase in their respective pathogenic mechanisms, which is not without

adverse consequences on the liver, and this can increase the morbidity and mortality as a result of complications arising from liver injuries.”

Author Response: Dear Reviewer, thank you for your remarks.

The authors are grateful to the reviewers for their valuable contributions which significantly improved the quality of this work. We very much hope the revised manuscript is accepted for publication in your Journal. Thank you for your consideration.

Sincerely yours,

Corresponding author

---

## [Decision Letter · Decision Letter 1]

15 Apr 2025

Serological evidence and factors associated to liver damage in malaria-typhoid infected patients consulting in two health facilities, Yaoundé-Cameroon

PONE-D-25-05617R1

Dear Dr. KOUAM,

We’re pleased to inform you that your manuscript has been judged scientifically suitable for publication and will be formally accepted for publication once it meets all outstanding technical requirements.

Kind regards,

Hope Onohuean, PhD

Academic Editor

PLOS ONE

Additional Editor Comments (optional):

Reviewers' comments:

Reviewer's Responses to Questions

**Comments to the Author**

Reviewer #1: All comments have been addressed

Reviewer #2: All comments have been addressed

2. Is the manuscript technically sound, and do the data support the conclusions?

Reviewer #1: Yes

Reviewer #2: Yes

3. Has the statistical analysis been performed appropriately and rigorously?

Reviewer #1: Yes

Reviewer #2: Yes

4. Have the authors made all data underlying the findings in their manuscript fully available?

Reviewer #1: Yes

Reviewer #2: Yes

5. Is the manuscript presented in an intelligible fashion and written in standard English?

Reviewer #1: Yes

Reviewer #2: Yes

Reviewer #1: The paper presents useful information for the prevention and contro of two diseases (malaria abd typhoid fever with common dymtoms. The authors have addressed alk my comments satisfactorily. Therefore I recommend the paper for publication in PLoS ONE journal.

Reviewer #2: The authors have thoroughly addressed the previously raised concerns. The manuscript is nearly ready for publication; however, I recommend the addition of a visual representation illustrating the connection between the infectious cycles of malaria and typhoid fever, particularly highlighting their interaction at the hepatic stage. With this final inclusion, the paper will be suitable for publication.

**Do you want your identity to be public for this peer review?** For information about this choice, including consent withdrawal, please see our Privacy Policy

Reviewer #1: No

Reviewer #2: **Yes: ** Ayomikun Kade

---

## [Editor Report · Acceptance letter]

PONE-D-25-05617R1

PLOS ONE

Dear Dr. KOUAM,

I'm pleased to inform you that your manuscript has been deemed suitable for publication in PLOS ONE. Congratulations! Your manuscript is now being handed over to our production team.

Kind regards,

on behalf of

Dr. Hope Onohuean

Academic Editor

PLOS ONE